# TAROT: Test-driven and Capability-adaptive Curriculum Reinforcement Fine-Tuning for Code Generation

## Abstract

Large Language Models (LLMs) are fundamentally changing the coding paradigm, known as vibe coding, yet synthesizing algorithmically sophisticated and robust code still remains a critical challenge. Incentivizing the deep reasoning capabilities of LLMs is essential to overcome this hurdle. Reinforcement Fine-Tuning (RFT) has emerged as a promising strategy to address this need. However, most existing approaches overlook the heterogeneous difficulty and granularity inherent in test cases, leading to an imbalanced distribution of reward signals and consequently biased gradient updates during training. To address this, we propose "TAROT", Test-driven and cApability-adaptive cuRriculum reinfOrcement fine-Tuning. TAROT systematically constructs, for each problem, a four-tier test suite (basic, intermediate, complex, edge), providing a controlled difficulty landscape for curriculum design and evaluation. Crucially, TAROT decouples curriculum progression from raw reward scores, enabling capability-conditioned evaluation and principled selection from a portfolio of curriculum policies rather than incidental test-case difficulty composition. This design fosters stable optimization and more efficient competency acquisition. Extensive experimental results reveal that the optimal curriculum for reinforcement fine-tuning in code generation is closely tied to a model's inherent capability, with less capable models achieving greater gains with an easy-to-hard progression, whereas more competent models excel under a hard-first curriculum. TAROT provides a reproducible method that adaptively tailors curriculum design to a model's capability, thereby consistently improving the functional correctness and robustness of the generated code. All code and data are released to foster reproducibility and advance community research at `https://anonymous.4open.science/r/TAROT-B675/`.

## 1 Introduction

Large Language Models (LLMs) are driving significant changes in software engineering, with automated code generation emerging as a pivotal application (Du et al., 2024; Jiang et al., 2024). Foundational models exhibit a strong capacity to translate natural language specifications into functional code, promising significant enhancements in developer productivity (Weber et al., 2024). Nevertheless, advancing the frontier toward synthesizing algorithmically sophisticated and highly robust solutions remains a critical challenge (Zhuo et al., 2025). The next key step hinges on significantly augmenting the deep reasoning and problem-solving faculties of these models.

Curriculum Learning (CL), a methodology that structures training data by difficulty (Bengio et al., 2009), presents a promising avenue for cultivating these capabilities and improving training efficiency. However, existing applications of CL in code generation primarily focus on sequencing entire problems based on coarse difficulty metrics (Naïr et al., 2024; Khant et al., 2025). While this inter-problem curriculum is intuitive, it neglects the nuanced, intra-problem difficulty gradient inherent in software verification. Human developers naturally employ practices like Test-Driven Development (TDD) (Beck, 2003), incrementally refining a solution against increasingly complex test cases to ensure robustness. Yet, this natural curriculum axis remains largely untapped in LLM training. Furthermore, reliance on problem-level sequencing often leads to flat reward landscapes when integrated with Reinforcement Fine-Tuning (RFT), dampening the learning signal. This oversight

of heterogeneous test-case difficulty results in imbalanced reward signals and consequently biased gradient updates during training, hindering the model's ability to acquire robust, sophisticated reasoning skills.

Furthermore, while the trend in curriculum learning for LLMs is shifting towards more dynamic approaches that progressively increase task complexity (Xu et al., 2024; Cheng et al., 2025), these methods predominantly define difficulty based on the intrinsic properties of the data or the task structure. For instance, curricula are often structured using automated metrics of the source code itself, such as cyclomatic complexity (Naïr et al., 2024), or by decomposing a problem into a fixed sequence of simpler subtasks (Dou et al., 2024). This prevailing focus on the data, rather than the learner, overlooks the crucial variable of the model's own evolving and multi-faceted capability. A curriculum tailored to an early-stage model may cause learning stagnation for a more advanced one, while a curriculum designed for experts can overwhelm a less-capable model and hinder its convergence. Therefore, for a more holistic approach to effective learning, curriculum design should consider not only the intrinsic properties of the data but also the evolving capabilities of the model itself, leading to a capability-adaptive framework.

To address these limitations, we introduce **TAROT**, a novel framework for **T**est-driven and c**A**pability-adaptive cu**R**riculum reinf**O**rcement fine-**T**uning. Crucially, TAROT decouples curriculum progression from raw reward scores, enabling capability-conditioned evaluation and principled selection from a portfolio of curriculum policies rather than incidental test-case difficulty composition. This design fosters stable optimization and more efficient competency acquisition. The framework's novelty is twofold. First, TAROT operationalizes the concept of an intra-problem difficulty gradient through a novel, test-driven curriculum. To instantiate this gradient, which is absent in standard coding datasets, we constructed the TAROT dataset. Each coding problem is systematically augmented with a test suite built upon four tiers of difficulty including basic, intermediate, complex, and edge cases. This structure defines difficulty as a spectrum of functional correctness. This engineered gradient directly counteracts the flat reward landscape common in RL, providing a structured and nuanced signal for learning robust solutions.

Second, we study capability-adaptive curriculum design. Given the TAROT dataset, we instantiate a portfolio of curriculum policies that vary along three axes, namely allocation across tiers, the sequence and proportion of tiers, and reward weighting across tiers. This setup enables capability-conditioned evaluation and principled selection among policies for models differing in effective capability influenced by model scale and specialization. Our central thesis is that the optimal learning path is capability dependent. In particular, less-capable models learn best with basic to complex progression, whereas more-capable models learn best by focusing on complex tiers.

To validate this thesis, this paper introduces the TAROT framework and demonstrate/s its effectiveness through comprehensive experiments, making the following primary contributions:

- A novel intra-problem, test driven curriculum for code generation, embodied in the new TAROT dataset. Each problem in our dataset is augmented with a four-tiered test suite (basic, intermediate, complex, edge cases) to provide a granular difficulty landscape and enable nuanced reward modeling.
- A capability conditioned study and guideline for curriculum design in code generation. We present a portfolio of curriculum policies in allocation sequence and reward weighting together with a reproducible evaluation protocol for capability conditioned comparison and principled selection informed by the characteristics of model capability such as scale and specialization.
- Comprehensive empirical validation demonstrating the effectiveness of TAROT. All models are fine-tuned using GRPO to leverage verifiable code execution rewards. Our experiments show that our capability-adaptive approach significantly improves model performance and training efficiency compared to the baselines.

## 2 RELATED WORKS

Our work is positioned at the intersection of two key research domains: curriculum learning for structuring pedagogical data, and reinforcement learning for policy optimization in code generation. We review prior work in these areas and highlight how TAROT offers a novel synthesis of both.

## 2.1 CURRICULUM LEARNING FOR CODE

Curriculum Learning (CL) is a training strategy inspired by human cognition that presents data to a model in a structured order, typically from simple to complex examples (Bengio et al., 2009). This method has been shown to accelerate convergence and improve generalization by guiding optimization toward better solutions. In the context of Large Language Models (LLMs), curricula have been implemented in various ways, such as using a teacher model to progressively generate more complex instructions, as seen in the Evol-Instruct method (Xu et al., 2024), or by fine-tuning on a small set of meticulously curated, high-quality examples as demonstrated by LIMA (Zhou et al., 2023). For code generation, where task complexity varies widely, CL is a particularly promising but challenging area. While many code datasets rely on manual difficulty labels, recent research has focused on more systematic approaches. A notable example is the use of automatic difficulty metrics, combining measures like cyclomatic complexity and Halstead difficulty, to sort problems into a multi-stage curriculum (Naïr et al., 2024). Training with this structured approach yielded significant gains, demonstrating the value of CL in the code domain. Other methods, like StepCoder, create an implicit curriculum by breaking a complex problem into a sequence of simpler code-completion subtasks (Dou et al., 2024). These efforts show a clear trend towards leveraging curricula to organize the training process for code generation. Our work contributes to this line of research by proposing a novel method to generate a tiered test suite that serves as the basis for our curriculum.

## 2.2 REINFORCEMENT FINE-TUNING FOR CODE LLMS

Reinforcement Learning (RL) is a dominant paradigm for aligning LLMs with desired behaviors, using techniques like RLHF (Ouyang et al., 2022), DPO (Rafailov et al., 2023), PPO (Schulman et al., 2017), GRPO (Shao et al., 2024), and GSPO (Zheng et al., 2025). In code generation, RL is adapted to optimize for functional correctness, typically using unit test outcomes as a reward signal. This "RL from unit test feedback" approach, while effective, often suffers from two key limitations: a sparse and "flat" reward landscape. The reward is sparse because a model gets no learning signal on complete failure, and it is flat because all successful solutions receive the same reward, regardless of the problem's difficulty. This flatness, where all successful solutions receive a similar reward regardless of the challenge, generates imbalanced reward signals that can lead to biased gradient updates. Recent work has begun to address these shortcomings. To combat sparse rewards, Process Reward Models have been introduced to provide dense, line-level feedback, guiding the model even when the final code is incorrect (Dai et al., 2025). To address the flat reward landscape, researchers are exploring ways to incorporate a sense of difficulty into the learning process. The idea of combining RL with a curriculum is gaining traction. For instance, some approaches use RL to guide a model through a curriculum of subtasks, while others dynamically adjust the curriculum during RL training using techniques like the Self-Evolving Curriculum, which treats problem selection as a multi-armed bandit problem to maximize learning progress (Chen et al., 2025). Our TAROT framework specifically addresses the flat reward problem by making the reward signal itself curriculum-aware. Instead of treating all successes equally, we modulate the reward based on the difficulty of the solved test tier, a concept inspired by curriculum design. By integrating this tiered reward scheme directly into a stable policy optimization algorithm, the TAROT framework provides a more nuanced learning gradient that encourages the model to master harder problems. This approach of infusing a static curriculum structure directly into the RL reward mechanism is a novel contribution that complements other recent innovations in the field.

## 3 TAROT FRAMEWORK

In this section, we elaborate on the details of the proposed TAROT framework which enhances the code generation capability of language models by having them solve test cases of varying difficulty in appropriate order and rewarding weights that are adaptively determined based on the model's baseline capability.

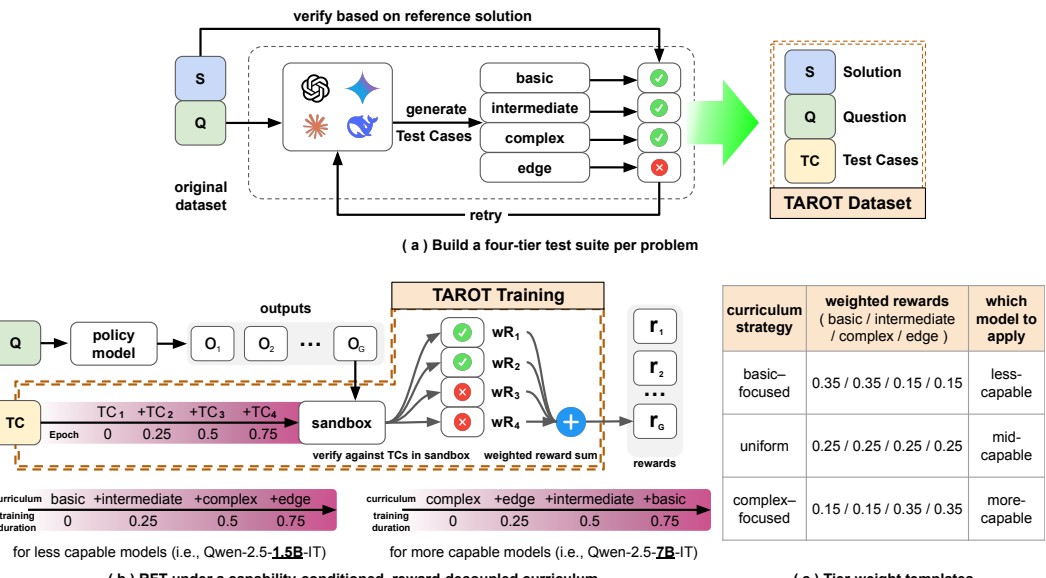

Figure 1: **Overview of TAROT framework.** (a) Build a four-tier test suite (basic/intermediate/-complex/edge) per problem using frontier LLMs and verify them against the reference solution. (b) Reinforcement fine-tuning under a capability-conditioned, reward-decoupled curriculum. Less capable models perform best with basic → complex, whereas more capable models perform best with complex → basic. (c) Tier-weight templates specifying reward weights for basic, intermediate, complex, and edge, with suggested use by capability buckets.

## 3.1 TAROT DATASET

A coding problem, denoted as P, is formally defined as a tuple consisting of three core components: a problem statement ($\mathcal{S}$), a reference solution ($\mathcal{R}$), and a set of test suite ($\mathcal{T}$). This relationship can be expressed as:

$$\mathcal{P} = (\mathcal{S}, \mathcal{R}, \mathcal{T}) \tag{1}$$

In this structure, the problem statement $\mathcal{S}$ outlines the task, and the reference solution $\mathcal{R}$ provides a correct implementation. The primary purpose of the test suite, $\mathcal{T}$, is to serve as a final validation mechanism to verify the correctness of a solver's proposed solution, but they are not designed to facilitate users' incremental learning processes. Consequently, the number and nature of test cases can vary significantly. For instance, a problem's test suite might consist of a single, simple case to verify the primary logic, or conversely, focus exclusively on complex edge cases, neither of which is structured to support a step-by-step learning process.

From a software engineering perspective, development is commonly test-driven. It begins with simple tests and progressively adds more complex and edge cases; implementations are refactored along the way, strengthening correctness and design. This staged expansion of the test suite mirrors the intuition behind curriculum learning. However, the test suite accompanying typical coding problems are not authored with this incremental pedagogy in mind. They are primarily designed for summative verification rather than stepwise scaffolding, so both their cardinality and difficulty mix vary widely and arbitrarily across problems.

To address the absence of a pedagogical structure, we introduce the TAROT dataset $\mathcal{D}_{\text{TAROT}}$, constructed by the procedure depicted in Figure 1 (a). Each problem in the dataset is augmented by incorporating a tiered test suite organized into four predefined difficulty levels (basic, intermediate, complex, and edge), without modifying the original statement or the reference solution. This structure is formally defined as follows:

$$\mathcal{D}_{\text{TAROT}} = \big\{ \big( \mathcal{S}_i,\ \mathcal{R}_i,\ \{\,\mathcal{T}_{i,l}\,\}_{l \in L} \big) \big\}_{i=1}^{N}, \tag{2}$$

$$L = \{\text{basic, intermediate, complex, edge}\}, \tag{3}$$

$$\mathcal{T}_i = \bigcup_{l \in L} \mathcal{T}_{i,l}, \tag{4}$$

$$\text{s.t.} \quad \forall\, i \in [N],\ \forall\, l \in L,\ \forall\, t \in \mathcal{T}_{i,l} :\ \text{Pass}(\mathcal{R}_i, t). \tag{5}$$

Here, $L$ is the set of difficulty levels, and the full suite for each problem is the union of its per-level subsets equation 4. By construction (see equation 5), every test case is validated against the reference solution, ensuring data quality. Any curriculum order (e.g., basic→complex or complex→basic) is imposed at training time and is not part of the dataset definition.

### 3.2 TAROT TRAINING MECHANISM

The TAROT training mechanism is designed to decouple the curriculum from raw test scores. It achieves this by utilizing two pre-defined components: a curriculum allocator that defines a fixed proportion of training focus for each difficulty tier $l \in L$, and tailored reward weights that prioritize tiers by placing greater value where the learning signal is most beneficial. During the training loop, the model first generates candidate solutions for a given problem. These solutions are then executed against the tiered test cases, and the resulting pass/fail outcomes are used to calculate and accumulate a tier-weighted return. Both the curriculum allocation and reward weights are aligned with the model's capability. The guiding principle is to concentrate the training signal within a zone of optimal difficulty, which is unique to each model's effective capability, a composite of instruction following fidelity and baseline coding proficiency. Therefore, the entire training schedule is pre-configured to match this profile, creating a fixed yet highly customized learning path.

Specifically, this means that models with lower baseline capability receive a larger share of basic and intermediate cases, whereas models with higher capability are assigned more complex and edge cases to push their frontier. The reward weights mirror this design, ensuring that successes on capability-appropriate tiers contribute more to the final objective.

We now formulate the training objective in the reinforcement learning setting. For each problem $P_i$ and difficulty level $l \in L$, we define the tier-level success of a policy $\pi$ as the average pass rate over that tier's tests.

$$r_{i,l}(\pi) \;=\; \frac{1}{|\mathcal{T}_{i,l}|} \sum_{t \in \mathcal{T}_{i,l}} \mathbf{1}\{\text{Pass}(\pi, t)\}. \tag{6}$$

Here, $\text{Pass}(\pi, t)$ indicates that the solution produced under $\pi$ satisfies test case $t$.

Given a curriculum allocation $\boldsymbol{\alpha} = (\alpha_l)_{l \in L}$ and reward weights $\boldsymbol{w} = (w_l)_{l \in L}$, we define the TAROT return for $P_i$ as a weighted sum over tiers.

$$R_{\text{TAROT}}\big(P_i, \pi; \boldsymbol{\alpha}, \boldsymbol{w}\big) \;=\; \sum_{l \in L} \alpha_l\, w_l\, r_{i,l}(\pi), \qquad \sum_{l \in L} \alpha_l = 1,\ \ w_l \geq 0. \tag{7}$$

We interpret $\alpha_l$ as the share of training effort assigned to tier $l$, and $w_l$ as how much a success on tier $l$ contributes given the model's baseline capability. Here, each $\alpha_l$ not only weights the contribution of tier $l$ to the return but also specifies the fraction of training updates allocated to that tier.

Training then maximizes the expected TAROT return over problems.

$$J_{\text{TAROT}}(\theta) \;=\; \mathbb{E}_{P_i \sim \mathcal{D}_{\text{TAROT}}}\Big[ R_{\text{TAROT}}\big(P_i, \pi_\theta; \boldsymbol{\alpha}, \boldsymbol{w}\big) \Big]. \tag{8}$$

This formulation provides a simple yet powerful objective function. By decoupling the allocation of training effort ($\boldsymbol{\alpha}$) from the valuation of success ($\boldsymbol{w}$), the TAROT framework enables a fine-grained, capability-matched curriculum. This approach moves beyond imposing a single, model-agnostic learning path, instead concentrating the training signal on the most productive difficulty tiers for any given model.

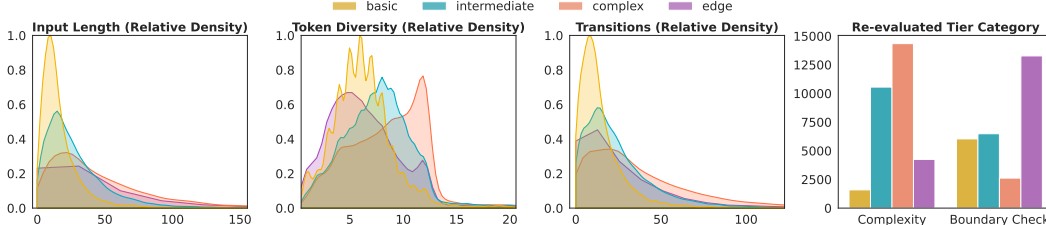

Figure 2: Quantitative and qualitative validation of the TAROT dataset. The KDE plots show the distribution of structural complexity, where the x-axis represents the metric's magnitude. Token Diversity (unique/total tokens) and Transitions (character class changes) serve as proxies for lexical and syntactic complexity, respectively. The systematic rightward shift confirms increasing difficulty across tiers. GPT-4o validation on the right confirms that complex tiers target algorithmic complexity, while edge tiers focus on boundary conditions.

Table 1: Overview of the experimental schedules for curriculum learning. Each strategy varies in reward distribution and the sequence of difficulties presented to the model. The abbreviations B, I, C, and E correspond to basic, intermediate, complex, and edge difficulty tiers, respectively. For staged curricula, transitions occur at 0.2, 0.4, and 0.6 of the total epoch.

| Strategy | Reward Weights (B, I, C, E) | Curriculum Schedule Progression |
| --- | --- | --- |
| Forward (Uniform) | (0.25, 0.25, 0.25, 0.25) | B → (B,I) → (B,I,C) → All |
| Forward (B & I Weighted) | (0.35, 0.35, 0.15, 0.15) | B → (B,I) → (B,I,C) → All |
| Forward (C & E Weighted) | (0.15, 0.15, 0.35, 0.35) | B → (B,I) → (B,I,C) → All |
| Reversed (C & E Weighted) | (0.15, 0.15, 0.35, 0.35) | C → (C,E) → (C,E,I) → All |
| Basic Only | (1.0, —, —, —) | Static |
| Complex Only | (—, —, 1.0, —) | Static |
| Edge Only | (—, —, —, 1.0) | Static |

## 4 EXPERIMENTS

### 4.1 EXPERIMENTAL SETTINGS

We construct a TAROT dataset based on 15k Python coding interview problems[1] with validated basic/intermediate/complex/edge test suites. As illustrated in Table 1, we design curriculum policies along two axes: allocation order and reward weighting. For allocation, we explore **Forward** (basic→edge), **Reversed** (edge→basic), and **Static** schedules. Transitions for staged curricula occur at 0.2, 0.4, and 0.6 of the total epoch. For weighting, we define three templates: **Uniform** (0.25 for all tiers), **B/I Weighted** (emphasizing the basic and intermediate tiers), and **C/Edge Weighted** (emphasizing the complex and edge tiers).

We evaluate TAROT training mechanism on a diverse suite of models, including Qwen2.5-Instruct, Qwen2.5-Coder-Instruct (1.5B, 3B, 7B) (Qwen et al., 2025; Hui et al., 2024), Gemma-2-IT (2B, 9B) (Team et al., 2024), and Qwen3-4B-Instruct-2507 (Yang et al., 2025) on well-known benchmarks including HumanEval (Chen et al., 2021), MBPP (Austin et al., 2021), HumanEval+, MBPP+ (Liu et al., 2024), LiveCodeBench v5 (Jain et al., 2024), CodeForces (Penedo et al., 2025), and CruxEval (Gu et al., 2024). This selection allows us to assess the framework's effectiveness across a wide spectrum of model scales, architectures, coding specializations, and performance tiers, including those at the frontier. All models are fine-tuned using GRPO (Shao et al., 2024). Full implementation details appear in Appendix B.

---

[1]https://huggingface.co/datasets/open-r1/verifiable-coding-problems-python

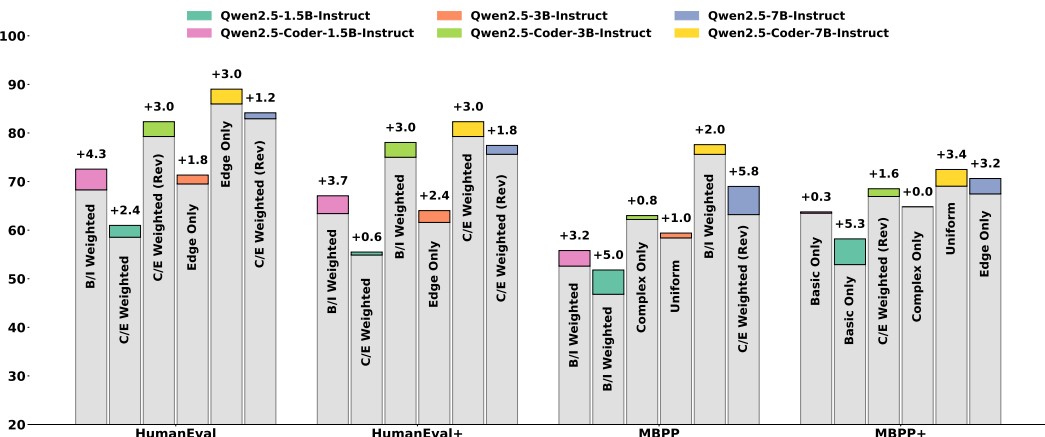

Figure 3: Experimental results for Qwen2.5-Instruct and Qwen2.5-Coder-Instruct on HumanEval, HumanEval+, MBPP, and MBPP+. Scores are pass@1. Numbers above bars indicate gains in percentage points relative to each model's base checkpoint. Labels inside bars indicate the best performing curriculum strategy.

## 4.2 EXPERIMENTAL RESULTS

To validate the empirical integrity of the TAROT dataset's tiering, we analyzed its structure using quantitative and qualitative metrics, as illustrated in Figure 2. The three KDE plots demonstrate a clear progression: as the tiers advance from basic to complex, the distributions for input length, token diversity, and character transitions all exhibit a consistent rightward shift, signifying a systematic increase in structural complexity. Furthermore, the qualitative bar chart reveals a crucial distinction between the two hardest tiers. It shows that test cases designed to probe complexity peak in the complex tier, while those targeting boundary checks are overwhelmingly concentrated in the edge tier. These complementary findings confirm that our four-level taxonomy not only stratifies overall difficulty but also effectively separates different types of challenge, establishing a robust foundation for the subsequent experiments.

Experimental results, illustrated in Figure 3, reveal a nuanced relationship between model scale, specialization, and optimal curriculum design. The Qwen2.5-Instruct models exhibit a straightforward, scale-dependent trend; the largest model (7B) performs best with complex-focused strategies, while the smallest (1.5B) benefits from a conventional basic-focused approach. However, this correlation with scale does not fully explain the performance. The coding specialized Qwen2.5-Coder models introduce a critical insight, as the mid-scale Qwen2.5-Coder-3B model displays a learning preference akin to the much larger Instruct-7B model despite its smaller parameter count. It achieves its peak HumanEval score using the same complex-focused strategy and decisively outperforms its general-purpose 3B counterpart. This finding strongly suggests that a model's prior specialization enhances its effective capability, making it a more critical determinant of the ideal learning path than parameter count alone.

To generalize these findings, the investigation was extended to the more recent Qwen3-4B-Instruct-2507. As detailed in Table 2, this newer model reinforces the core thesis. The optimal curriculum strategy, *C/E Weighted*, consistently outperforms the base model across all evaluated benchmarks, yielding substantial gains ranging from +2.12 to +4.26 percentage points. Notably, these improvements were achieved on a model that already possesses a strong performance baseline, confirming that curriculum learning is an effective method for eliciting further gains even from highly capable models. This result is significant because it demonstrates that a model's preference for advanced, complex-focused curricula is not merely a function of parameter count but is strongly tied to its overall capability. The success of this strategy on a powerful, state-of-the-art Qwen3-4B-Instruct-2507 further solidifies the argument that the most capable models benefit most from curricula that prioritize challenging examples.

Table 2: Performance comparison of curriculum strategies for Qwen3-4B-Instruct-2507, highlighting the top-performing *C/E Weighted* curriculum strategy against the base model. The performance delta is shown in percentage points.

| Strategy | HumanEval | HumanEval+ | MBPP | MBPP+ |
|---|---|---|---|---|
| Base | 89.02% | 78.66% | 52.60% | 56.61% |
| C/E Weighted | **91.46%** (+2.44pp) | **82.92%** (+4.26pp) | **55.20%** (+2.60pp) | **58.73%** (+2.12pp) |

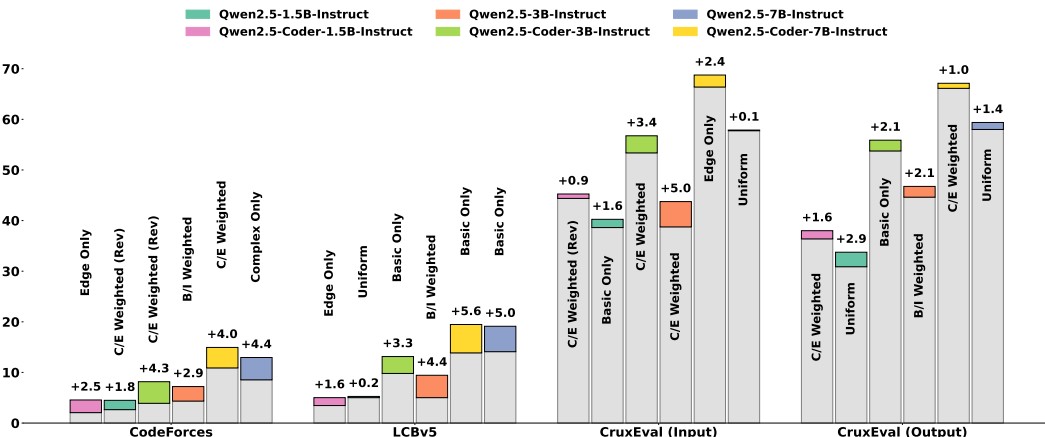

Figure 4: Experimental results for Qwen2.5-Instruct and Qwen2.5-Coder-Instruct models on Code-Forces, LiveCodeBench v5 (LCBv5), and CruxEval. Scores are the overall accuracy across easy, medium, and hard problems. Numbers above bars indicate gains in percentage points relative to each model's base checkpoint. Labels inside bars indicate the best performing curriculum strategy.

We attribute this divergence to the "Zone of Optimal Difficulty". For more-capable models, basic tier is trivial and yield negligible learning signal, whereas complex tier provides the high-entropy signal needed for improvement. Conversely, less-capable models facing Complex tiers initially suffer from sparse rewards, leading to training collapse. Therefore, we recommend using basic-focused curricula for less-capable models to ensure stability, while adopting complex-focused curricula for more-capable or code-specialized models to maximize gradient efficiency.

## 4.3 IN-DEPTH ANALYSIS

**Out-Of-Distribution Benchmarks** Evaluations on OOD benchmarks including CodeForces, LiveCodeBench v5, and CruxEval reveal that while TAROT consistently outperforms baselines, the optimal curriculum strategy is highly task-dependent rather than universal (Figure 4). For instance, Qwen2.5-7B achieved peak performance with the *Basic Only* curriculum on LiveCodeBench v5, whereas the *C/E Weighted* strategy proved most effective for CruxEval and CodeForces. This divergence stems from varying skill alignments between coding interview-style training data and downstream tasks. Consequently, effective curriculum selection must account for the target domain's computational structure, necessitating future research into task-specific intra-problem test design and adaptive policy selection.

**Comparison with Standard Reward Schemes** To verify that the performance gains of TAROT stem from its capability-adaptive curriculum strategy, we compare our framework against standard reward shaping strategies commonly used in reinforcement learning for code generation. These baselines utilize the full four-tier test suite throughout the training. Specifically, we evaluate two standard RL baselines: **Avg-reward**, where the reward is calculated as the average pass rate across the four tiers ($R \in [0, 1]$), and **Pass@All**, which represents a strict functional correctness setting

Table 3: Performance comparison between TAROT and Standard RL Baselines. TAROT outperforms conventional reward schemes that use the same test cases but lack curriculum scheduling. Highest scores are highlighted in **bold**.

| Model | Strategy | HumanEval | HumanEval+ | MBPP | MBPP+ | CodeForces | LCBv5 | CruxEval |
|-------|----------|-----------|------------|------|-------|------------|-------|----------|
| **Qwen2.5-1.5B-Instruct** | | | | | | | | |
| | Avg-reward | 59.15% | 54.27% | 49.20% | 57.93% | 2.72% | 3.70% | 38.75/32.87% |
| | Pass@All | **60.98%** | **56.10%** | 44.60% | 53.43% | 2.72% | 3.94% | 34.62/31.75% |
| | TAROT (Best) | **60.98%** | 55.49% | **51.80%** | **58.20%** | **4.49%** | **5.26%** | **40.20/33.75%** |
| **Qwen2.5-7B-Instruct** | | | | | | | | |
| | Avg-reward | 83.75% | 76.22% | 66.00% | 69.84% | 11.41% | 11.95% | 56.37/58.50% |
| | Pass@All | 81.10% | 73.78% | 63.00% | 68.52% | 9.49% | 14.81% | 55.62/56.63% |
| | TAROT (Best) | **84.15%** | **77.44%** | **69.00%** | **70.63%** | **12.95%** | **19.12%** | **57.88/59.38%** |

where a reward of 1 is assigned only if all four tiers pass, and 0 otherwise ($R \in \{0, 1\}$). As detailed in Table 3, TAROT consistently outperforms both standard reward schemes across all benchmarks.

**Architectural Generalization**  Applying TAROT to Gemma2 architecture validated our hypothesis that baseline proficiency, not parameter count, dictates the optimal learning path. For the larger Gemma2-9B-IT, *Complex Only* curriculum offered no decisive advantage, as simpler strategies like *Basic Only* often proved superior on key benchmarks shown in Table 4. This principle was even more starkly visible with the smaller Gemma2-2B-IT. As detailed in Appendx F, most curricula were actively harmful, leading to a performance collapse from a sparse reward signal. In stark contrast, a *Basic Only* strategy focused on fundamentals yielded substantial improvements. This demonstrates that for less-capable models, a fundamentals-first curriculum is a prerequisite for successful fine-tuning, whereas unstructured approaches can be severely detrimental.

Table 4: Performance comparison for Gemma2-9B-IT across key curriculum strategies. Highest scores on each benchmark are highlighted in **bold**.

| Strategy | HumanEval | HumanEval+ | MBPP | MBPP+ | CodeForces | LCBv5 | CruxEval |
|----------|-----------|------------|------|-------|------------|-------|----------|
| Base | 60.37% | 54.88% | 59.60% | 65.08% | 8.61% | 11.83% | 45.63/47.63% |
| Uniform | **65.85%** | 57.93% | 59.20% | 64.55% | 10.82% | 13.62% | **51.63**/47.13% |
| Basic Only | 63.41% | 56.10% | **60.40%** | 65.08% | 9.49% | **14.70%** | 51.00/48.00% |
| Complex Only | **65.85%** | **60.37%** | 58.60% | 64.55% | 9.93% | 12.54% | 48.25/48.00% |

For completeness, we report the full per-strategy and per-curriculum results for all Qwen2.5-Instruct, Qwen2.5-Coder-Instruct, and Qwen3-4B-Instruct models HumanEval, MBPP, and the OOD benchmarks in Appendix G, Tables 8 and 7. We further analyze the sensitivity to training hyper-parameters (temperature, GRPO $\beta$) and the inference-time maximum token limit in Appendix D and E, and investigate the training dynamics and reward correlations in Appendix H.

## 5 CONCLUSION

We introduced TAROT, a test-driven and capability-adaptive framework for curriculum reinforcement fine-tuning in code generation. TAROT moves beyond the conventional one-size-fits-all approach by constructing a four-tier, intra-problem test suite that allows curriculum design to be tailored to a model's unique abilities. Experiments confirmed our central thesis that the optimal learning path is capability-dependent: less-capable models benefit most from an basic-focused progression, while more-capable models excel with curricula that prioritize complex-focused challenges. We found that the most critical factor is not parameter count alone but a more holistic effective capability, which accounts for a model's prior specialization. Ultimately, TAROT provides a practical framework for enhancing the code generation capabilities of large language models. Our framework proved its value across a wide spectrum of models, from less-capable base models to highly proficient code-specialized and state-of-the-art foundation models, confirming its broad applicability. While significant, our results also show that the optimal curriculum is task-dependent, pointing toward future work in developing domain-specific test suites and automated policy selection methods.

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

# A  TEST CASE GENERATION PROMPTS

To ensure the consistent generation of high-quality, four-tiered test cases, we designed a detailed prompt template. This template, listed in Table 5, guides the language model to act as an expert software engineer and produce test cases that adhere to our specific difficulty criteria.

Table 5: The Prompt template used to generate a tiered test suite per a given coding problem. The problem statement and the default test case from the original source is injected into {problem_statement} and {baseline_test_case} placeholders respectively.

---

You are an expert software engineer with extensive experience in designing comprehensive unit tests. Your task is to generate four distinct unit tests for a given code implementation based solely on the provided problem statement. Treat this as a black-box testing exercise—focus exclusively on the inputs and expected outputs without assuming any details about the internal implementation.

**Important:** A baseline test case will be provided separately. Each test case you generate must be more challenging than the baseline test case.

Please generate four unit tests with the following guidelines:

1. **Basic Complexity Test (label as "basic"):**

   - Use simple, straightforward inputs.
   - Validate the core behavior under normal conditions.
   - Focus on the happy path scenario.
   - This should be the least challenging test case relative to the others.

2. **Medium Complexity Test (label as "intermediate"):**

   - Include moderately complex inputs with some edge conditions.
   - Test with mixed data types or larger inputs.
   - Incorporate common edge cases and boundary values.
   - Ensure this test is more challenging than the basic test.

3. **High Complexity Test (label as "complex"):**

   - Use complex, nested, or structured inputs.
   - Validate advanced functionality and complex logic paths.
   - Stress test the implementation with challenging scenarios.
   - This test should be more intricate than both the basic and intermediate tests.

4. **Edge Case Test (label as "edge"):**

   - Use extreme boundary conditions and special cases.
   - Validate behavior with empty, null, or invalid inputs.
   - Focus on error handling and exception scenarios.
   - This should be the most challenging test case among the four.

For each test case, follow the JSON format provided in the example below (include only the input and expected output):

```
{
  "language": "python",
  "test_cases": [
    {
      "input": "4\n4\n0001\n1000\n0011\n0111\n3\n010\n101\n0\n2\n00000\n00001\n4\n01\n001\n0001\n00001\n",
      "output": "1\n3 \n-1\n0\n\n2\n1 2 \n",
      "type": "stdin_stdout",
      "label": "basic",
      "reason": "This test represents simple, straightforward input conditions."
    }
  ]
}
```

Remember:

- Do not assume any knowledge about the internal code; base your tests purely on the input-output behavior described in the problem statement.

- Ensure that each of your test cases is incrementally more challenging than the baseline test case provided.

**Problem Statement:** {problem_statement}

**Baseline Test Case:** {baseline_test_case}

---

# B IMPLEMENTATION DETAILS

**TAROT Dataset** Our experiments utilize the TAROT dataset, which we constructed by augmenting approximately 15,000 problems from the verifiable-coding-problems-python dataset[2]. For each problem, we employed OpenAI's the most powerful o3 and o4 models[3] with the highest reasoning effort to generate a four-tiered test suite with distinct levels: basic, intermediate, complex, and edge. The specific prompts used for this generation process are detailed in Appendix A. To ensure high quality, every generated test case was validated against the reference solution, and any problem with even one failing tier was discarded. This rigorous curation process yields a final dataset of approximately 60,000 tiered test suites (15,000 problems × 4 tiers). Samples of these generated tiered test cases can be found in Appendix I.

**Model Selection** To validate our methodology, we selected a diverse set of models to investigate four key research questions: (1) the effect of model scale, to test our hypothesis that the optimal curriculum is capability-dependent, using three Qwen2.5 models of varying sizes (1.5B, 3B, 7B) (Qwen et al., 2025); (2) the impact of specialization, to determine if TAROT can further enhance models already proficient in coding, using their code-specialized counterparts (Hui et al., 2024); (3) architectural generalizability, to test if our findings apply beyond a single model family, by incorporating two instruction-tuned Gemma2 models (2B, 9B) (Team et al., 2024); and (4) pushing performance frontiers, to assess if our framework can improve even state-of-the-art models with strong baselines, by fine-tuning the recent Qwen3-4B-Instruct-2507 (Yang et al., 2025). For all models, we used their instruction-tuned variants to ensure a foundational code-generation capability, a prerequisite for effective RL-based fine-tuning.

**Training Details** We fine-tune all selected models for a single epoch using the TAROT framework. For policy optimization, we employ GRPO (Shao et al., 2024). All models are trained using the AdamW optimizer with a constant learning rate of $1 \times 10^{-6}$. We set the global batch size to 8, reducing it to 4 for larger models (Qwen2.5-7B-Instruct, Qwen2.5-Coder-7B-Instruct, and Gemma2-9B-IT) to accommodate memory constraints. The maximum input and completion token lengths were set to 1,024 and 4,096, respectively. For GRPO-specific settings, we generated 8 candidate completions per prompt to estimate the policy advantage, with the core hyperparameter $\beta$ set to 0.01 in our main experiments. We provide an ablation study on key training hyperparameters, including the GRPO $\beta$ value (0.1, 0.05, 0.01) and the sampling temperature during training (1.0, 0.7, 0.5), in Appendix D.

The GRPO hyperparameter $\beta$ controls the strength of the Kullback-Leibler (KL) divergence regularization term, which penalizes the policy for deviating too far from the original base model's behavior. The training temperature, in turn, manages the exploration-exploitation trade-off; higher values encourage the model to sample a wider variety of solutions (exploration), while lower values cause it to refine high-probability ones (exploitation). Our ablation studies were designed to identify the optimal settings for these crucial parameters within our code generation task.

All fine-tuning experiments were conducted on a server with 8 x NVIDIA A100 (80 GB) GPUs, running CUDA 12.4 and PyTorch 2.6. Our implementation is based on open-source libraries including Transformers (Wolf et al., 2020), TRL (von Werra et al., 2020), vLLM (Kwon et al., 2023), and Open-R1 (Hugging Face, 2025).

**Evaluation Metrics** We evaluate the efficacy of the TAROT framework on a comprehensive suite of code generation benchmarks. For functional correctness, we measure the pass@1 metric on HumanEval (Chen et al., 2021), MBPP (Austin et al., 2021), and their more challenging variants, HumanEval+ and MBPP+ (Liu et al., 2024). To assess competitive problem-solving skills, we use the overall accuracy on LiveCodeBench v5 (Jain et al., 2024) and CodeForces (Penedo et al., 2025), averaged across their difficulty tiers. Finally, the model's code reasoning capability is evaluated using the input and output prediction accuracy on CruxEval (Gu et al., 2024). The detailed generation parameters and execution environment are described in Appendix C.

---

[2]https://huggingface.co/datasets/open-r1/verifiable-coding-problems-python
[3]https://platform.openai.com/docs/models

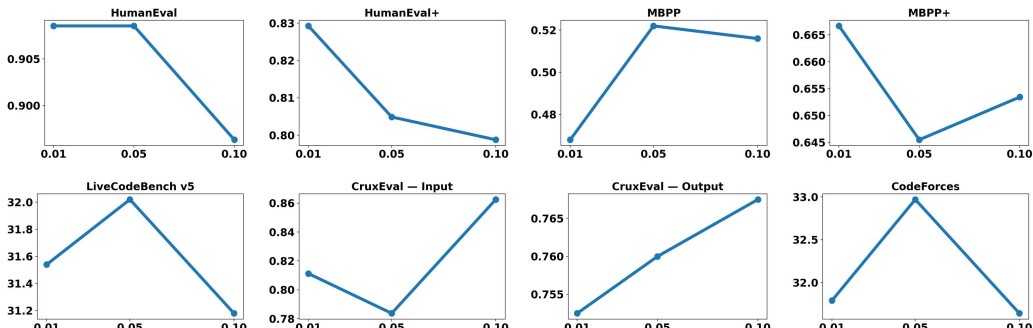

Figure 5: Performance sensitivity to the GRPO hyperparameter $\beta$. The plots show the final pass@1 or accuracy scores on various benchmarks as $\beta$ is varied. The optimal value is task-dependent; for instance, HumanEval and HumanEval+ benefit from a smaller $\beta$ (0.01) that allows greater policy exploration, whereas MBPP and CodeForces achieve peak performance with a larger $\beta$ (0.05) that enforces stronger regularization.

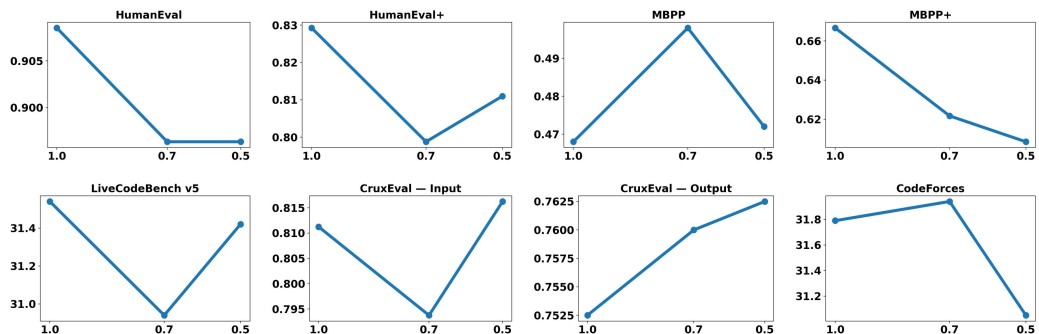

Figure 6: Performance sensitivity to the sampling temperature during training. The plots illustrate the final benchmark scores for different training temperatures. A higher temperature of 1.0, which encourages greater exploration, is optimal for benchmarks like HumanEval and HumanEval+. In contrast, other benchmarks such as MBPP show a preference for a more moderate temperature of 0.7, highlighting that the ideal exploration-exploitation balance is task-specific.

## C  GENERATION AND EXECUTION ENVIRONMENT

The entire evaluation pipeline is managed by the EvalChemy framework (Raoof et al., 2025). We follow the benchmark-specific generation configurations predefined within the framework—such as temperature, top-p, prompt formatting, and stopping criteria—to ensure consistency with established evaluation protocols. By default, the maximum completion tokens for each benchmark adhered to its standard setting; however, for an ablation study on generation length (Appendix E), we systematically increased this limit to 4,096, 8,192, and 16,384 tokens to observe performance trends.

All code generation for evaluation was conducted by serving the fine-tuned models via the vLLM framework (Kwon et al., 2023) on servers equipped with 4 x NVIDIA A100 (80 GB) GPUs, using a batch size of 64. The resulting code is executed in a secure, sandboxed Python 3.11 environment, where a strict 10-second timeout is enforced for each test case to prevent infinite loops and manage evaluation time.

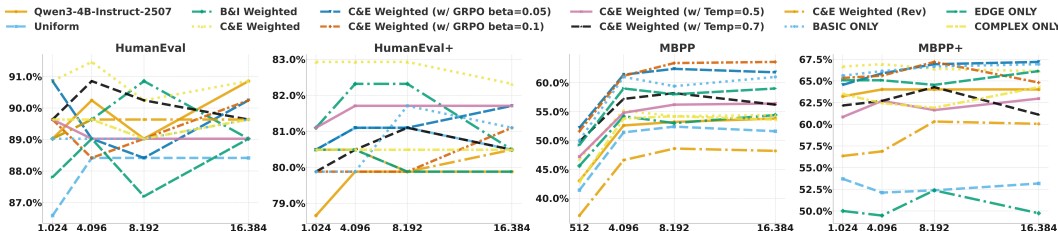

Figure 7: Performance sensitivity to the maximum completion token limit at inference time for Qwen3-4B-Instruct-2507 fine-tuned on various curriculum strategies. The results reveal a clear, benchmark-dependent dichotomy. For function-completion tasks like HumanEval and HumanEval+, performance tends to degrade as the token limit increases beyond 4,096, suggesting that a larger generation space may encourage verbose, error-prone solutions. Conversely, for benchmarks like MBPP and MBPP+, a larger token limit is generally beneficial, indicating that their problem structures may require more extensive code to solve correctly.

## D  HYPERPARAMETER SENSITIVITY ANALYSIS

This section provides ablation studies on two key training hyperparameters to analyze their impact on final benchmark performance: the GRPO regularization coefficient $\beta$ and the sampling temperature during training.

**Impact of GRPO's $\beta$**  The hyperparameter $\beta$ in GRPO controls the strength of the Kullback-Leibler (KL) divergence regularization, which prevents the fine-tuned policy from deviating excessively from the original base model. The results of varying $\beta$ are shown in Figure 5. Performance sensitivity to $\beta$ is not uniform across benchmarks. For function-synthesis tasks like HumanEval and HumanEval+, a small $\beta$ of 0.01, which allows for greater policy exploration, yields the best results. Conversely, benchmarks like MBPP and CodeForces appear to benefit from slightly stronger regularization ($\beta = 0.05$). This variance suggests that the optimal regularization strength is task-dependent. We selected $\beta = 0.01$ for our main experiments as it proved most effective on our primary evaluation benchmarks.

**Impact of Training Temperature**  The sampling temperature manages the exploration-exploitation trade-off during training. The results, presented in Figure 6, indicate that a higher temperature of 1.0, which encourages greater exploration of diverse solutions, is optimal for HumanEval and HumanEval+. However, other benchmarks show different trends; MBPP, for example, peaks at a more conservative temperature of 0.7. This highlights that the optimal degree of exploration is also task-specific, and suggests that task-adaptive temperature scheduling could be a potential area for future work.

## E  IMPACT OF MAXIMUM COMPLETION TOKENS AT INFERENCE TIME

We analyzed the impact of the maximum completion token limit during inference on the fine-tuned Qwen3-4B model, with results presented in Figure 7. The findings reveal a clear, benchmark-dependent dichotomy. On function-completion tasks like HumanEval and HumanEval+, performance generally degrades as the token limit increases beyond 4,096. In stark contrast, benchmarks like MBPP and MBPP+ benefit from a larger generation space, with optimal results often found at 8,192 or 16,384 tokens.

This divergence suggests that for tasks requiring concise solutions, such as those in HumanEval, a larger token limit may encourage verbose and error-prone code. Conversely, the nature of MBPP problems may necessitate a longer generation process to fully develop the correct logic. This analysis underscores a critical point for standardized evaluation: the ideal setting for maximum completion tokens is highly contingent on the characteristics of the target benchmark.

## F ADDITIONAL RESULTS ON GEMMA2-2B-IT

This appendix provides the full curriculum comparison for Gemma2-2B-IT as in Table 6. Unlike larger or stronger models, Gemma2-2B-IT exhibits curriculum fragility: most curricula depress performance, consistent with the observation in the main text that sparse reward signals can cause collapse for less-capable models. In contrast, *Basic Only*—a fundamentals-first schedule—yields the most reliable gains among the tested strategies.

These results reinforce our capability-dependent view of curriculum design: for weaker models, emphasizing simpler tiers is a prerequisite for successful fine-tuning, whereas complex-focused or mixed curricula can be harmful.

Table 6: Performance comparison for Gemma2-2B-IT across all curriculum strategies. Scores are colored and bolded based on their deviation from the Base strategy (blue for higher, red for lower).

| Strategy | HumanEval | HumanEval+ | MBPP | MBPP+ | CodeForces | LCBv5 | CruxEval |
|---|---|---|---|---|---|---|---|
| Base | 42.07% | 34.76% | 41.20% | 47.09% | 2.21% | 4.30% | 37.50/26.88% |
| Uniform | **39.02**% | **31.09**% | **33.80**% | **39.95**% | **0.22**% | **3.58**% | **33.00/26.63**% |
| B/I Weighted | **35.98**% | **32.32**% | **35.60**% | **42.06**% | **0.22**% | 4.30% | **38.63/27.25**% |
| C/E Weighted | **41.46**% | **34.15**% | **39.20**% | **48.41**% | **0.22**% | **3.94**% | **36.63/26.75**% |
| C/E Weighted (Rev) | **40.86**% | **35.37**% | **40.20**% | **44.44**% | **0.44**% | **3.94**% | **35.63/26.75**% |
| Basic Only | **44.51**% | **37.20**% | **38.60**% | **46.83**% | **1.77**% | **3.94**% | **39.88/27.88**% |
| Edge Only | **39.63**% | **35.37**% | **38.00**% | **46.03**% | **0.22**% | 4.30% | **39.00/28.13**% |
| Complex Only | 42.07% | **36.59**% | **37.00**% | **45.77**% | 2.21% | **2.87**% | **35.63/27.55**% |

## G FULL BENCHMARK TABLES (QWEN2.5 & QWEN3-4B)

We report the complete benchmark results for all curriculum strategies on Qwen2.5 family (1.5B/3B/7B, including Coder variants) and Qwen3-4B-Instruct-2507. These tables expand the main figures by listing pass@1 on HumanEval, HumanEval+, MBPP, MBPP+, and average accuracy of CodeForces, LiveCodeBench v5, and CruxEval for every strategy in Table 8 and 7.

Consistent with the main text, the *C/E Weighted* strategy tends to be the top performer for the more-capable Qwen3-4B model, improving over the base across all four code-function benchmarks. The full per-strategy breakdowns here allow exact comparison across OOD benchmarks as well.

## H TRAINING DYNAMICS ANALYSIS

**The Limits of the Reward Signal.** Figure 8 (a) shows that the training reward increases stably and is clearly separated by model capacity, indicating a stable optimization process. Note that initial rewards are relatively low even for capable models; this is due to strict output formatting

Table 7: Comprehensive performance evaluation of all curriculum strategies on Qwen3-4B-Instruct-2507. The highest score on each benchmark is highlighted in **bold**. The performance of Qwen3-Coder-30B-A3B-Instruct is included to enable comparison against a leading code-specialized model.

| Model | Strategy | HumanEval | HumanEval+ | MBPP | MBPP+ | CodeForces | LCBv5 | CruxEval |
|---|---|---|---|---|---|---|---|---|
| **Qwen3-Coder-30B-A3B-Instruct** | | | | | | | | |
| | Base | 94.51% | 86.59% | 73.80% | 75.13% | 29.65% | 37.63% | 81.75/79.25% |
| **Qwen3-4B-Instruct-2507** | | | | | | | | |
| | Base | 89.02% | 78.66% | 52.60% | 56.61% | 33.63% | 32.02% | 78.25/**77.75**% |
| | Uniform | 88.41% | 80.09% | 35.30% | 53.70% | 31.86% | 31.96% | 79.37/75.75% |
| | B/I Weighted | 89.63% | 81.09% | 28.00% | 52.38% | 33.04% | **33.81**% | 79.50/75.38% |
| | C/E Weighted | **91.46**% | **82.92**% | **55.20**% | **58.73**% | 31.79% | 31.54% | **81.12**/75.25% |
| | C/E Weighted (Rev) | 89.63% | 80.48% | 36.20% | 35.98% | **34.66**% | 31.66% | 79.50/76.00% |
| | Basic Only | 89.63% | 79.87% | 39.80% | 56.34% | 33.11% | 31.90% | 78.50/75.00% |
| | Edge Only | 89.63% | 79.88% | 47.20% | 56.61% | 31.86% | 30.59% | 80.25/74.00% |
| | Complex Only | 90.85% | 80.48% | 28.60% | 51.85% | 30.61% | 31.30% | 80.37/76.37% |

Table 8: Comprehensive performance evaluation of all curriculum strategies across Qwen2.5-Instruct and Qwen2.5-Coder-Instruct models (1.5B, 3B, 7B). For each model size, the highest score on each benchmark is highlighted in **bold**.

| Model | Strategy | HumanEval | HumanEval+ | MBPP | MBPP+ | CodeForces | LCBv5 | CruxEval |
|---|---|---|---|---|---|---|---|---|
| **Qwen2.5-7B-Instruct** | | | | | | | | |
| | Base | 82.93% | 75.61% | 63.20% | 67.46% | 8.54% | 14.10% | 57.75/58.00% |
| | Uniform | 82.93% | 73.78% | 67.40% | 67.46% | 12.36% | 14.93% | **57.88/59.38%** |
| | B/I Weighted | 78.05% | 76.83% | 67.60% | 69.58% | 11.56% | 15.89% | 57.25/59.25% |
| | C/E Weighted | 79.27% | 73.78% | 66.20% | 70.37% | 10.89% | 15.77% | 57.13/55.50% |
| | C/E Weighted (Rev) | **84.15%** | **77.44%** | 69.00% | 70.11% | 8.24% | 15.41% | 57.88/56.38% |
| | Basic Only | 82.32% | 75.61% | 66.20% | 68.52% | 12.29% | **19.12%** | 55.63/57.50% |
| | Edge Only | 83.54% | 76.22% | 67.60% | **70.63%** | 11.11% | 17.08% | 56.13/57.75% |
| | Complex Only | **84.15%** | 75.61% | **69.00%** | 69.05% | **12.95%** | 17.80% | 57.25/56.38% |
| **Qwen2.5-3B-Instruct** | | | | | | | | |
| | Base | 69.51% | 61.59% | 58.40% | **64.81%** | 4.34% | 5.02% | 38.75/44.63% |
| | Uniform | **71.34%** | 63.41% | **59.40%** | 63.49% | 6.92% | 8.72% | 42.00/42.50% |
| | B/I Weighted | 69.51% | 62.20% | 59.00% | 63.49% | **7.21%** | **9.44%** | 42.38/**46.75%** |
| | C/E Weighted | 69.51% | 62.80% | 56.60% | 63.76% | **7.21%** | 7.17% | **43.75**/44.50% |
| | C/E Weighted (Rev) | 70.12% | 62.80% | 57.00% | 63.49% | 6.92% | 8.00% | 43.63/42.50% |
| | Basic Only | 66.46% | 59.15% | **59.40%** | 64.02% | 6.33% | 6.09% | 40.50/44.13% |
| | Edge Only | **71.34%** | **64.02%** | 58.20% | 62.70% | 6.11% | 7.05% | 43.13/42.63% |
| | Complex Only | 67.68% | 60.37% | 59.00% | **64.81%** | 6.84% | 6.33% | 41.25/42.88% |
| **Qwen2.5-1.5B-Instruct** | | | | | | | | |
| | Base | 58.54% | 54.88% | 46.80% | 52.91% | 2.65% | 5.02% | 38.63/30.88% |
| | Uniform | **60.98%** | 54.88% | 50.00% | 57.14% | 3.68% | **5.26%** | 37.13/**33.75%** |
| | B/I Weighted | 59.15% | 54.27% | **51.80%** | 57.94% | 3.83% | 4.54% | 36.00/29.75% |
| | C/E Weighted | **60.98%** | **55.49%** | 49.40% | 56.08% | 3.61% | 5.02% | 34.75/32.38% |
| | C/E Weighted (Rev) | 56.71% | 52.44% | 50.40% | **58.20%** | **4.49%** | 4.90% | 34.00/31.75% |
| | Basic Only | 57.32% | 53.05% | 50.60% | **58.20%** | 4.05% | 4.66% | **40.25/33.00%** |
| | Edge Only | 55.49% | 50.61% | 50.20% | 56.08% | 3.75% | 4.42% | 35.50/31.50% |
| | Complex Only | 59.76% | 54.88% | **51.80%** | 55.29% | 3.46% | 4.54% | 36.13/33.38% |
| **Qwen2.5-Coder-7B-Instruct** | | | | | | | | |
| | Base | 85.98% | 79.27% | 75.60% | 69.05% | 10.89% | 13.86% | 66.38/66.13% |
| | Uniform | 85.76% | 79.27% | 77.20% | **72.49%** | 13.98% | 17.68% | 66.50/66.38% |
| | B/I Weighted | 84.76% | 78.66% | **77.60%** | 71.96% | 13.32% | 17.44% | 68.38/65.88% |
| | C/E Weighted | 87.80% | **82.32%** | 76.20% | 70.90% | **14.94%** | 19.24% | 66.25/**67.13%** |
| | C/E Weighted (Rev) | **88.41%** | 81.10% | 75.00% | 71.42% | 13.98% | 19.12% | 68.63/65.00% |
| | Basic Only | 85.98% | 79.88% | 76.20% | 71.96% | 14.86% | **19.47%** | 67.50/66.38% |
| | Edge Only | 79.02% | 81.07% | 77.20% | 71.96% | 12.14% | 19.12% | **68.75**/66.00% |
| | Complex Only | 87.80% | 80.49% | 76.60% | 70.90% | 14.35% | 18.16% | 67.75/66.50% |
| **Qwen2.5-Coder-3B-Instruct** | | | | | | | | |
| | Base | 79.27% | 75.00% | 62.20% | 66.93% | 3.90% | 9.80% | 53.38/53.75% |
| | Uniform | 81.10% | 76.83% | 62.00% | 67.20% | 7.21% | 10.75% | 54.00/54.75% |
| | B/I Weighted | 81.71% | **78.05%** | 61.40% | 66.93% | 6.70% | 9.80% | 54.25/53.38% |
| | C/E Weighted | 79.88% | 76.83% | 61.00% | 67.46% | 8.02% | 10.51% | **56.75**/53.50% |
| | C/E Weighted (Rev) | **82.32%** | 77.44% | 62.00% | **68.52%** | **8.17%** | 10.75% | 52.63/55.13% |
| | Basic Only | 80.49% | 76.22% | 62.80% | 66.67% | 7.95% | **13.14%** | 55.88/**55.88%** |
| | Edge Only | 79.27% | 75.00% | 62.60% | 66.14% | 7.21% | 10.63% | 53.75/53.25% |
| | Complex Only | 78.05% | 73.78% | **63.00%** | 67.72% | 7.65% | 10.63% | 53.13/55.25% |
| **Qwen2.5-Coder-1.5B-Instruct** | | | | | | | | |
| | Base | 68.29% | 63.41% | 52.60% | 63.49% | 2.06% | 3.46% | 44.38/36.38% |
| | Uniform | 71.34% | 65.24% | 52.80% | 62.96% | 4.56% | 4.42% | 44.75/36.38% |
| | B/I Weighted | **72.56%** | 64.02% | **55.80%** | 62.70% | 4.19% | 4.66% | 45.13/35.75% |
| | C/E Weighted | 71.34% | 66.65% | 54.60% | 62.96% | 3.46% | 4.42% | 45.13/**38.00%** |
| | C/E Weighted (Rev) | **72.56%** | 64.20% | 54.20% | 62.96% | 3.38% | 4.18% | **45.25**/37.00% |
| | Basic Only | 70.12% | 64.02% | 54.00% | **64.76%** | 4.49% | 4.66% | 43.25/36.00% |
| | Edge Only | **72.56%** | **67.10%** | 53.60% | 62.17% | **4.56%** | **5.02%** | 44.86/35.63% |
| | Complex Only | 71.34% | 66.46% | 53.20% | 63.49% | 3.31% | 4.54% | 43.75/37.36% |

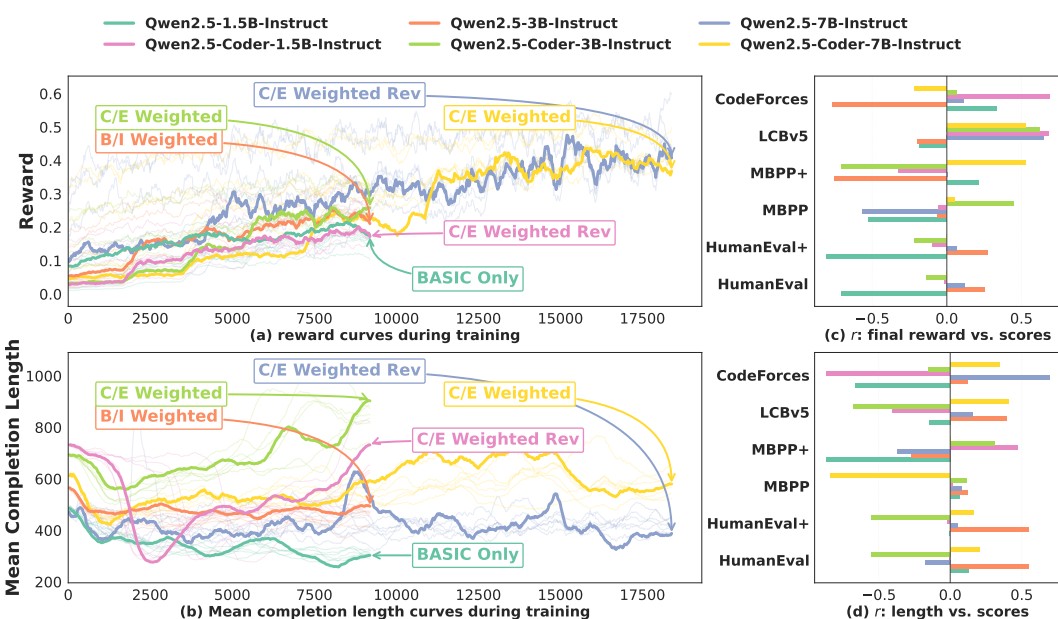

Figure 8: Training dynamics vs. downstream performance. (a) and (b) show the reward and the mean completion length curves during reinforcement fine-tuning, and the annotations mark the curriculum strategy with the best average downstream performance. (c) and (d) show the Pearson correlation coefficient $r$ of the final rewards vs. benchmark scores and the mean completion length vs. benchmark scores, respectively. Light, semi-transparent lines represent alternative curriculum strategies, while the solid, annotated lines correspond to the best-performing strategy for each model. Some trajectories terminate earlier than others because different model sizes utilize varying batch sizes and gradient accumulation steps under a fixed total compute budget.

requirements and execution timeouts enforced by the sandbox, which the models quickly adapt to during the early stages of fine-tuning. This pattern suggests that the policy learns the training distribution well and that stronger models achieve higher reward levels under the same curriculum. However, the reward observed during training does not reliably anticipate downstream benchmark outcomes. As shown in Figure 8 (c), the final reward has only a weak Pearson correlation coefficient with benchmark scores, which means that runs with similar rewards can still deliver very different levels of task performance.

**Conciseness as a Proxy for Advanced Reasoning.** A different perspective comes from analyzing completion length. Figure 8 (b) shows that models with greater capability tend to produce shorter solutions as training progresses, and this tendency becomes more pronounced for stronger configurations. Importantly, Figure 8 (d) indicates that mean completion length exhibits a stronger negative correlation with benchmark scores than the reward does, implying that conciseness aligns better with final solution quality. Shorter programs are more likely to capture the essential reasoning steps without unnecessary detours, whereas longer outputs often reflect uncertainty or inefficient search. These observations support using solution conciseness as a practical secondary proxy for advanced reasoning quality, complementing the reward based perspective and providing a more informative early indicator of downstream performance.

## I  SAMPLE TIERED TEST CASES

Table 9- 18 present concrete examples of the four-tiered test cases generated for several problems in the TAROT dataset. These samples illustrate a clear and intentional progression in difficulty and scope, which is a cornerstone of our framework.

The tiers are generally designed to validate different aspects of a solution. Basic tiers focus on the core logic of a problem with simple, straightforward inputs. Following this, intermediate and complex tiers introduce greater difficulty through larger inputs, more intricate scenarios, or patterns requiring more sophisticated algorithmic reasoning. Finally, edge tiers are designed to test for robustness by probing boundary conditions, constraints, and performance-intensive cases such as large numbers or long strings. This tiered structure exemplifies the intra-problem difficulty gradient that forms the basis of our capability-adaptive curriculum.

Table 9: A sample from TAROT dataset comprising 4-tiered test cases: basic, intermediate, complex, and edge. The Reason column details the rationale for each tier assignment.

```
Solve the following coding problem using the programming language python:

You are given two positive integer numbers a and b. Permute (change order) of
the digits of a to construct maximal number not exceeding b. No number in input
and/or output can start with the digit 0.

It is allowed to leave a as it is.

Input
The first line contains integer a (1 ≤ a ≤ 10^18). The second line contains integer
b (1 ≤ b ≤ 10^18). Numbers don't have leading zeroes. It is guaranteed that answer
exists.

Output
Print the maximum possible number that is a permutation of digits of a and is
not greater than b. The answer can't have any leading zeroes. It is guaranteed
that the answer exists. The number in the output should have exactly the same
length as number a. It should be a permutation of digits of a.

Examples
Input
123 222

Output
213

Input
3921 10000

Output
9321

Input
4940 5000

Output
4940

The input will be given via stdin and the output should be printed to stdout by
your code.

Now solve the problem by providing the code.
```

| Test cases | Basic | Intermediate | Complex | Edge |
|---|---|---|---|---|
| **Input** | 21 | 3051 | 98761230 | 111222333444555666 |
| | 12 | 5310 | 98765000 | 1000000000000000000 |
| **Output** | 12 | 5310 | 98763210 | 666555444333222111 |
| **Reason** | A simple 2-digit case where swapping the digits yields the only valid permutation ≤ bound, illustrating the happy path. | A 4-digit case including zero, requiring the algorithm to match the upper bound exactly with a permutation of the digits. | An 8-digit case where matching the bound fails at a later position, forcing backtracking and a maximal tail fill. | An extreme boundary case with an 18-digit input and a longer 19-digit bound, where any valid permutation fits, so the result is the digits sorted in descending order. |

Table 10: A sample from TAROT dataset comprising 4-tiered test cases: basic, intermediate, complex, and edge. The Reason column details the rationale for each tier assignment.

```
Solve the following coding problem using the programming language python:

Winter is here at the North and the White Walkers are close. John Snow has an
army consisting of n soldiers. While the rest of the world is fighting for the
Iron Throne, he is going to get ready for the attack of the White Walkers.

He has created a method to know how strong his army is. Let the i-th soldier's
strength be a_i. For some k, we call the indices i_1, i_2, ..., i_k a clan if i_1  <  i_2  <
...   <   i_k and gcd(a_{i_1}, a_{i_2}, ..., a_{i_k})   >   1. The strength of that clan is defined as
k   ·   gcd(a_{i_1}, a_{i_2}, ..., a_{i_k}). The strength of the army is defined by the sum of the
strengths of all possible clans.

Your task is to find the strength of his army. As the number may be very large,
you have to print it modulo 1000000007 (10^9 + 7).

Greatest common divisor (gcd) of a sequence of integers is the maximum possible
integer so that each element of the sequence is divisible by it.

-----Input-----
The first line contains integer n (1  ≤  n  ≤  200000) − the size of the army. The
second line contains n integers a_1, a_2, ..., a_n (1  ≤  a_i  ≤  1000000) − denoting the
strengths of his soldiers.

-----Output-----
Print one integer − the strength of John Snow's army modulo 1000000007 (10^9 + 7).

-----Examples-----
Input
3
3 3 1

Output
12

Input
4
2 3 4 6

Output
39

-----Note-----
In the first sample the clans are {1}, {2}, {1, 2} so the answer will be 1·3+1·3+2·3 =
12

The input will be stdin and you should print your solution to stdout

Now solve the problem and return the code.
```

| Test cases | Basic | Intermediate | Complex | Edge |
|---|---|---|---|---|
| **Input** | 4 | 6 | 7 | 5 |
| | 2 3 5 7 | 2 4 8 3 9 6 | 2 2 2 2 2 2 2 | 1 1 1 1 1 |
| **Output** | 17 | 119 | 896 | 0 |
| **Reason** | All strengths are prime, so only single-soldier clans contribute. | Mix of primes and composites yields clans of various sizes and gcds. | Uniform strengths where every nonempty subset is a valid clan (gcd=2). | All strengths are 1, so no clan has gcd¿1; result is zero. |

Table 11: A sample from TAROT dataset comprising 4-tiered test cases: basic, intermediate, complex, and edge. The Reason column details the rationale for each tier assignment.

```
Solve the following coding problem using the programming language python:

A permutation — is a sequence of length n integers from 1 to n, in which all the numbers
occur exactly once. For example, [1], [3, 5, 2, 1, 4], [1, 3, 2] — permutations, and [2, 3,
2], [4, 3, 1], [0] — no.

Polycarp was recently gifted a permutation a[1 ...n] of length n. Polycarp likes trees more
than permutations, so he wants to transform permutation a into a rooted binary tree. He
transforms an array of different integers into a tree as follows:

    • The maximum element of the array becomes the root of the tree;
    • All elements to the left of the maximum — form a left subtree (which is built
      according to the same rules but applied to the left part of the array), but if there
      are no elements to the left of the maximum, then the root has no left child;
    • All elements to the right of the maximum — form a right subtree (which is built
      according to the same rules but applied to the right side of the array), but if there
      are no elements to the right of the maximum, then the root has no right child.

For example, if he builds a tree by permutation a = [3, 5, 2, 1, 4], then the root will
be the element a_2 = 5, and the left subtree will be the tree that will be built for the
subarray a[1 ...1] = [3], and the right one — for the subarray a[3 ...5] = [2, 1, 4]. As a
result, the following tree will be built:

<image> The tree corresponding to the permutation a=[3, 5, 2, 1, 4].

Another example: let the permutation be a=[1, 3, 2, 7, 5, 6, 4]. In this case, the tree
looks like this:

<image> The tree corresponding to the permutation a=[1, 3, 2, 7, 5, 6, 4].

Let us denote by d_v the depth of the vertex a_v, that is, the number of edges on the path
from the root to the vertex numbered a_v. Note that the root depth is zero. Given the
permutation a, for each vertex, find the value of d_v.

Input

The first line contains one integer t (1 ≤ t ≤ 100) — the number of test cases. Then t test
cases follow. The first line of each test case contains an integer n (1 ≤ n ≤ 100) — the
length of the permutation. This is followed by n numbers a_1, a_2, ..., a_n — permutation a.

Output

For each test case, output n values — d_1, d_2, ..., d_n.

Example

Input
3
5
3 5 2 1 4
1
1
4
4 3 1 2

Output
1 0 2 3 1
0
0 1 3 2

The input will be stdin and you should print your solution to stdout

Now solve the problem and return the code.
```

| Test cases | Basic | Intermediate | Complex | Edge |
|---|---|---|---|---|
| **Input** | 1
3
1 2 3 | 2
4
2 1 4 3
5
5 4 3 2 1 | 1
10
3 8 2 5 10 9 1 7
4 6 | 1
15
1 2 3 4 5 6 7 8
9 10 11 12 13 14
15 |
| **Output** | 2 1 0 | 1 2 0 1
0 1 2 3 4 | 2 1 3 2 0 1 3 2
4 3 | 14 13 12 11 10 9
8 7 6 5 4 3 2 1
0 |
| **Reason** | Simple ascending permutation forming a left-skewed tree under normal conditions. | Includes a mixed permutation and a strictly decreasing permutation to test right-skewed tree and boundary values. | Complex permutation of length 10 to test multiple recursion levels and both left and right subtrees. | Maximum ascending chain of length 15 to test deep recursion and large boundary condition. |

Table 12: A sample from TAROT dataset comprising 4-tiered test cases: basic, intermediate, complex, and edge. The Reason column details the rationale for each tier assignment.

```
Solve the following coding problem using the programming language python:

Princess'Marriage

Marriage of a princess

English text is not available in this practice contest.

A brave princess in a poor country, knowing that gambling payouts are determined by the parimutuel
method, felt more familiar with gambling and was convinced of her victory in gambling. As a
result, he spent more money than ever before and lost enough to lose all the taxes paid by
the people. The King, who took this situation seriously, decided to marry the princess to the
neighboring country. By doing this, I thought that I would like the princess to reflect on her
daily activities and at the same time deepen her friendship with neighboring countries and receive
financial assistance.

The princess and the prince of the neighboring country liked each other, and the kings of both
countries agreed on a political marriage. The princess triumphantly went to the neighboring
country with a little money in her hand. On the other hand, the motive for the princess to marry
is to pursue the unilateral interests of the king, and the aide of the prince of the neighboring
country who thinks that it is not pleasant shoots countless thugs along the way to make the
princess dead. It was.

The path the princess will take has already been decided. There are a total of L post stations
on the path of the princess. For convenience, the departure and arrival points are also set as
post stations, and each post station is called S1, S2, ...SL. The princess shall be in S1 first,
visit the post station in ascending order (S2, S3 ...in that order), and finally go to SL. At
the post station, you can pay money to hire an escort, and as long as you have the money, you
can contract for as long as you like to protect the princess. The cost of hiring an escort is 1
gold per distance. Note that the princess can also partially protect the section through which she
passes. The distance between Si and Si + 1 is given by Di, and the expected value of the number of
times a thug is attacked per distance between Si and Si + 1 is given by Pi.

Find the expected number of thugs to reach your destination when the princess has a budget of M
and hires an escort to minimize the expected number of thugs.

Input
The input consists of multiple datasets. Each dataset has the following format.

> N M
> D1 P1
> D2 P2
> ...
> DN PN

Two integers are given in the first row of each dataset, representing the number of intervals N
(1 ≤ N ≤ 10,000 and the budget M (0 ≤ M ≤ 1,000,000,000 of the princess difference, respectively.
The next N lines show information about the path the princess takes. Each line contains two
integers, and the i-th line is the expected value of the interval Di (1 ≤ Di ≤ 10,000) and the
number of attacks when moving one unit distance between them Pi (0 ≤ Pi ≤ 10) ). The end of the
input is represented by a data set with N = 0 and M = 0. Do not output the calculation result for
this data set.

Output
For each dataset, output the expected number of times the princess will be attacked by thugs to
your destination.

Sample Input
2 8
5 6
4 5
3 1
5 10
5 10
5 10
0 0

Output for the Sample Input
Five
140

The input will be given via stdin and the output should be printed to stdout by your code.
```

| Test cases | Basic | Intermediate | Complex | Edge |
|---|---|---|---|---|
| **Input** | 4 7 | 6 12 | 10 30 | 3 0 |
| | 3 2 | 5 3 | 10 1 | 5 2 |
| | 4 1 | 2 0 | 5 5 | 10 4 |
| | 1 5 | 7 2 | 8 5 | 7 3 |
| | 2 2 | 3 3 | 6 3 | 4 100 |
| | 0 0 | 4 1 | 12 2 | 5 2 |
| | | 6 2 | 4 5 | 10 4 |
| | | 0 0 | 7 3 | 7 3 |
| | | | 9 4 | 8 0 |
| | | | 3 0 | 2 1000 |
| | | | 11 2 | 100 0 |
| | | | 0 0 | 200 0 |
| | | | | 0 0 |
| **Output** | 3 | 22 | 83 | 71 |
| | | | | 0 |
| | | | | 0 |
| **Reason** | Simple scenario with multiple segments and straightforward positive Pi values; tests basic greedy coverage under a limited budget. | Moderate number of segments including Pi=0, ensuring segments with no attacks are ignored and budget partially covers higher-Pi segments. | Larger set of segments with ties in Pi values and varied distances, requiring correct sorting and partial coverage among equal-Pi segments. | Edge conditions including zero budget, budget exceeding total distance, and segments with Pi=0 to verify no-protection and full-protection behaviors. |

Table 13: A sample from TAROT dataset comprising 4-tiered test cases: basic, intermediate, complex, and edge. The Reason column details the rationale for each tier assignment.

```
Solve the following coding problem using the programming language python:

Valera loves his garden, where n fruit trees grow.

This year he will enjoy a great harvest! On the i-th tree b_i fruit grow, they will ripen
on a day number a_i. Unfortunately, the fruit on the tree get withered, so they can only be
collected on day a_i and day a_i  +  1 (all fruits that are not collected in these two days,
become unfit to eat).

Valera is not very fast, but there are some positive points. Valera is ready to work every
day. In one day, Valera can collect no more than v fruits. The fruits may be either from
the same tree, or from different ones. What is the maximum amount of fruit Valera can
collect for all time, if he operates optimally well?

-----Input-----
The first line contains two space-separated integers n and v (1 ≤ n,v ≤ 3000) — the number
of fruit trees in the garden and the number of fruits that Valera can collect in a day.

Next n lines contain the description of trees in the garden. The i-th line contains two
space-separated integers a_i and b_i (1 ≤ a_i,b_i ≤ 3000) — the day the fruits ripen on the i-th
tree and the number of fruits on the i-th tree.

-----Output-----
Print a single integer — the maximum number of fruit that Valera can collect.

-----Examples-----
Input
2 3
1 5
2 3

Output
8

Input
5 10
3 20
2 20
1 20
4 20
5 20

Output
60

-----Note-----
In the first sample, in order to obtain the optimal answer, you should act as follows.
On the first day collect 3 fruits from the 1-st tree. On the second day collect 1 fruit
from the 2-nd tree and 2 fruits from the 1-st tree. On the third day collect the remaining
fruits from the 2-nd tree.

In the second sample, you can only collect 60 fruits, the remaining fruit will simply
wither.

The input will be stdin and you should print your solution to stdout

Now solve the problem and return the code.
```

| Test cases | Basic | Intermediate | Complex | Edge |
|---|---|---|---|---|
| **Input** | 2 5 | 3 1 | 5 5 | 2 1000 |
| | 1 3 | 1 2 | 1 4 | 2999 1500 |
| | 3 4 | 2 2 | 2 6 | 3000 2500 |
| | | 3 2 | 2 3 | |
| | | | 5 10 | |
| | | | 6 2 | |
| **Output** | 7 | 4 | 25 | 3000 |
| **Reason** | No overlapping ripening days and capacity exceeds daily fruits; collect all fruits on their ripening days. | Capacity is only 1 per day with overlapping two-day windows; requires optimal scheduling across consecutive days. | Multiple trees ripen on the same days, gaps between ripening days, and moderate capacity to stress multi-day planning. | Ripening on the maximum allowed days (2999 and 3000) tests boundary handling and two-day collection windows at the end of the range. |

Table 14: A sample from TAROT dataset comprising 4-tiered test cases: basic, intermediate, complex, and edge. The Reason column details the rationale for each tier assignment.

```
Solve the following coding problem using the programming language python:

Bessie the cow has just intercepted a text that Farmer John sent to Burger Queen!
However, Bessie is sure that there is a secret message hidden inside.

The text is a string s of lowercase Latin letters. She considers a string t as
hidden in string s if t exists as a subsequence of s whose indices form an arithmetic
progression. For example, the string aab is hidden in string aaabb because it occurs
at indices 1, 3, and 5, which form an arithmetic progression with a common difference
of 2. Bessie thinks that any hidden string that occurs the most times is the secret
message. Two occurrences of a subsequence of S are distinct if the sets of indices
are different. Help her find the number of occurrences of the secret message!

For example, in the string aaabb, a is hidden 3 times, b is hidden 2 times, ab is
hidden 6 times, aa is hidden 3 times, bb is hidden 1 time, aab is hidden 2 times, aaa
is hidden 1 time, abb is hidden 1 time, aaab is hidden 1 time, aabb is hidden 1 time,
and aaabb is hidden 1 time. The number of occurrences of the secret message is 6.

-----Input-----

The first line contains a string s of lowercase Latin letters (1 ≤ |s| ≤ 10^5) | the text
that Bessie intercepted.

-----Output-----

Output a single integer — the number of occurrences of the secret message.

-----Examples-----
Input
aaabb

Output
6

Input
usaco

Output
1

Input
lol

Output
2

-----Note-----

In the first example, these are all the hidden strings and their indice sets: a
occurs at (1), (2), (3) b occurs at (4), (5) ab occurs at (1,4), (1,5), (2,4), (2,5), (3,4),
(3,5) aa occurs at (1,2), (1,3), (2,3) bb occurs at (4,5) aab occurs at (1,3,5), (2,3,4)
aaa occurs at (1,2,3) abb occurs at (3,4,5) aaab occurs at (1,2,3,4) aabb occurs at
(2,3,4,5) aaabb occurs at (1,2,3,4,5) Note that all the sets of indices are arithmetic
progressions.

In the second example, no hidden string occurs more than once.

In the third example, the hidden string is the letter l.

The input will be stdin and you should print your solution to stdout

Now solve the problem and return the code.
```

| Test cases | Basic | Intermediate | Complex | Edge |
|---|---|---|---|---|
| **Input** | abab | abacaba | abababab | z |
| **Output** | 3 | 6 | 10 | 1 |
| **Reason** | A simple alternating pattern to validate basic subsequence counting. | Mixed letters and repeating patterns to test moderately complex subsequences. | Longer alternating pattern to stress test counting of many arithmetic-progression subsequences. | Minimal input length boundary case. |

Table 15: A sample from TAROT dataset comprising 4-tiered test cases: basic, intermediate, complex, and edge. The Reason column details the rationale for each tier assignment.

```
Solve the following coding problem using the programming language python:

The ZCO scholarship contest offers scholarships to first time ZCO participants. You are participating in
it for the first time. So you want to know the number of participants who'll get the scholarship. You know
that the maximum number of scholarships offered is R and there are a total of N participants numbered
from 1 to N. Out of these, you know the set of people (denoted by X) who you know, had participated in
previous year ZCOs and hence, they shall not get the scholarship. Further, as the world isn't free from
plagiarism, so is the case with the scholarship contest. And from your secret sources, you also know
the set of people (denoted by set Y) who were involved in plagiarism and therefore aren't eligible for
scholarship either.

Find out the number of participants who shall get the scholarship.

PS: Don't ask how so many scholarships are being offered when you see the constraints on R. You never
questioned it when in mathematics classes, some person bought 80 watermelons twice just to compare them
and save 1.

-----Input:-----
- The first line will contain a single integer, T, the number of testcases. Then the testcases follow.
- The first line of each test case contains four integers; N, R, |X| and |Y| denoting the number of
participants, maximum number of scholarships offered, number of old participants, and the number of
participants involved in plagiarism, respectively.
- The second line of each test case contains |X| space separated integers x_1, x_2 ... x_|X| denoting the
indices of people who participated in previous years. If X is empty, this line is skipped and no empty
line is in the input.
- The third line of each test case contains |Y| space separated integers y_1, y_2 ... y_|Y| denoting the indices
of people who are involved in plagiarism. If Y is empty, this line is skipped and no empty line is in
input.

-----Output:-----
For each testcase, print a single integer in a new line, denoting the number of participants who shall get
the scholarship.

-----Constraints-----
- 1 ≤ T ≤ 1000
- 1 ≤ N ≤ 10^15
- 0 ≤ R ≤ 10^15
- 0 ≤ |X|,|Y| ≤ min(N, 2 * 10^5)
- 1 ≤ x_i, y_i ≤ N
- All x_i are distinct
- All y_i are distinct
- Sum of |X| over all test cases does not exceed 5 * 10^5
- Sum of |Y| over all test cases does not exceed 5 * 10^5

-----Subtasks-----
- 20 points : 1 ≤ N ≤ 10^3, and the sum of N over all test cases does not exceed 3 * 10^3
- 30 points : 1 ≤ N ≤ 2 * 10^5, and the sum of N over all test cases does not exceed 5 * 10^5
- 50 points: Original constraints

-----Sample Input:-----
3
5 3 0 1
4
10 2 4 6
3 1 7 6
4 3 1 5 9 7
10 4 4 6
3 1 7 6
4 3 1 5 9 7

-----Sample Output:-----
3
2
3

-----EXPLANATION:-----
- In the first testcase, only participant 4 is involved in plagiarism, and thus not eligible for the
scholarship. No user has participated in previous years, and so no empty line is there in the sample. All
participants except participant 4 are eligible for the scholarship, but only three of them get it because
R = 3.
- Both second and third testcases are the same, except for R. In both samples, only participants 2, 8 and
10 are eligible for scholarships.
- In the second testcase, since the maximum number of scholarships is 2, only 2 participants get
scholarships.
- In the third testcase, all three eligible participants get scholarships.

The input will be stdin and you should print your solution to stdout
```

| Test cases | Basic | Intermediate | Complex | Edge |
|---|---|---|---|---|
| **Input** | 1
4 2 1 1
1
4 | 2
7 4 3 2
2 4 5
4 6
5 1 0 2
3 5 | 3
1000000000000
1000000000000 2 3
1 1000000000000
500000000000 1
999999999999
20 15 5 5
1 2 3 4 5
4 5 6 7 8
50 100 3 2
10 20 30
30 40 | 2
1000000000000000 0
0 0
5 10 3 3
1 2 3
3 4 5 |
| **Output** | 2 | 3
1 | 999999999996
12
46 | 0
0 |
| **Reason** | Basic test with a single test case, non-empty X and Y sets without overlap, validating core functionality. | Medium complexity with multiple test cases, overlapping X and Y in the first, and an empty X set in the second. | High complexity with very large N and R values, moderate X and Y sizes, and multiple test cases to stress-test the implementation. | Edge case with maximum boundary values and zero scholarships in the first, and X and Y covering all participants in the second, testing empty sets and full exclusion. |

Table 16: A sample from TAROT dataset comprising 4-tiered test cases: basic, intermediate, complex, and edge. The Reason column details the rationale for each tier assignment.

```
Solve the following coding problem using the programming language python:

Polycarp has recently got himself a new job. He now earns so much that his old wallet can't even store all
the money he has. Berland bills somehow come in lots of different sizes. However, all of them are shaped as
rectangles (possibly squares). All wallets are also produced in form of rectangles (possibly squares).
```

A bill $x \times y$ fits into some wallet $h \times w$ if either $x \le h$ and $y \le w$ or $y \le h$ and $x \le w$. Bills can overlap with each other in a wallet and an infinite amount of bills can fit into a wallet. That implies that all the bills Polycarp currently have fit into a wallet if every single one of them fits into it independently of the others.

```
Now you are asked to perform the queries of two types:
```

$+$ $x$ $y$ | Polycarp earns a bill of size $x$ $\times$ $y$; $?$ $h$ $w$ | Polycarp wants to check if all the bills he has earned to this moment fit into a wallet of size $h \times w$.

It is guaranteed that there is at least one query of type $1$ before the first query of type $2$ and that there is at least one query of type $2$ in the input data. For each query of type $2$ print "YES" if all the bills he has earned to this moment fit into a wallet of given size. Print "NO" otherwise.

```
-----Input-----
```

The first line contains a single integer $n$ ($2 \le n \le 5 \cdot 10^5$) | the number of queries.

Each of the next $n$ lines contains a query of one of these two types:

$+$ $x$ $y$ ($1 \le x, y \le 10^9$) | Polycarp earns a bill of size $x \times y$; $?$ $h$ $w$ ($1 \le h, w \le 10^9$) | Polycarp wants to check if all the bills he has earned to this moment fit into a wallet of size $h \times w$.

It is guaranteed that there is at least one query of type $1$ before the first query of type $2$ and that there is at least one query of type $2$ in the input data.

```
-----Output-----
```

For each query of type $2$ print "YES" if all the bills he has earned to this moment fit into a wallet of given size. Print "NO" otherwise.

```
-----Example-----
Input
9
+ 3 2
+ 2 3
? 1 20
? 3 3
? 2 3
+ 1 5
? 10 10
? 1 5
+ 1 1

Output
NO
YES
YES
YES
NO

-----Note-----
```

The queries of type $2$ of the example:

Neither bill fits; Both bills fit (just checking that you got that bills can overlap); Both bills fit (both bills are actually the same); All bills fit (too much of free space in a wallet is not a problem); Only bill $1 \times 5$ fit (all the others don't, thus it's "NO").

The input will be stdin and you should print your solution to stdout

| Test cases | Basic | Intermediate | Complex | Edge |
|---|---|---|---|---|
| **Input** | 4
+ 4 5
? 5 4
? 4 4
? 6 4 | 7
+ 2 7
+ 3 3
+ 7 2
? 3 7
? 4 6
+ 10 1
? 10 5 | 13
+ 5 5
+ 6 4
+ 9 1
? 5 5
? 9 1
? 4 9
+ 2 8
? 8 9
+ 7 7
? 7 7
? 8 7
? 10 8
? 6 6 | 8
+ 1000000000 1
+ 1 1000000000
+ 500000000
500000000
? 1000000000
1000000000
? 999999999
1000000000
? 1000000000
499999999
+ 1000000000
1000000000
? 1000000000
1000000000 |
| **Output** | YES
NO
YES | YES
NO
YES | NO
NO
NO
YES
NO
NO
YES
NO | YES
YES
NO
YES |
| **Reason** | Single bill with queries testing orientation and size validation under straightforward conditions. | Multiple bills including duplicates and interleaved adds and queries testing correct global dimension tracking. | Complex interleaving of many adds and queries with varying dimensions to stress test global maximum computations. | Extreme boundary values testing maximum limits and strict comparison edge where one dimension is just below requirement. |

Table 17: A sample from TAROT dataset comprising 4-tiered test cases: basic, intermediate, complex, and edge. The Reason column details the rationale for each tier assignment.

```
Solve the following coding problem using the programming language python:

Valera had an undirected connected graph without self-loops and multiple edges consisting
of n vertices. The graph had an interesting property: there were at most k edges adjacent
to each of its vertices. For convenience, we will assume that the graph vertices were
indexed by integers from 1 to n.

One day Valera counted the shortest distances from one of the graph vertices to all other
ones and wrote them out in array d.

Thus, element d[i] of the array shows the shortest distance from the vertex Valera chose to
vertex number i.

Then something irreparable terrible happened. Valera lost the initial graph. However, he
still has the array d. Help him restore the lost graph.

Input

The first line contains two space-separated integers n and k (1 ≤ k ≤ 105). Number n shows
the number of vertices in the original graph. Number k shows that at most k edges were
adjacent to each vertex in the original graph.

The second line contains space-separated integers d[1], d[2], ..., d[n] (0 ≤ d[i] < n). Number
d[i] shows the shortest distance from the vertex Valera chose to the vertex number i.

Output

If Valera made a mistake in his notes and the required graph doesn't exist, print in the
first line number -1. Otherwise, in the first line print integer m (0 ≤ m ≤ 106) — the
number of edges in the found graph.

In each of the next m lines print two space-separated integers ai and bi (1 ≤ ai, bi ≤
n; ai ≠ bi), denoting the edge that connects vertices with numbers ai and bi. The graph
shouldn't contain self-loops and multiple edges. If there are multiple possible answers,
print any of them.

Examples

Input
3 2
0 1 1

Output
3
1 2
1 3
3 2

Input
4 2
2 0 1 3

Output
3
1 3
1 4
2 3

Input
3 1
0 0 0

Output
-1

The input will be given via stdin and the output should be printed to stdout by your code.
```

| Test cases | Basic | Intermediate | Complex | Edge |
|---|---|---|---|---|
| **Input** | 4 2
0 1 1 2 | 7 3
0 1 2 2 1 2 3 | 10 3
0 1 1 1 2 2 2 2
2 3 | 5 3
0 2 2 3 3 |
| **Output** | 3
1 2
1 3
2 4 | 6
1 2
1 5
2 3
2 4
5 6
3 7 | 9
1 2
1 3
1 4
2 5
2 6
3 7
3 8
4 9
5 10 | -1 |
| **Reason** | Simple BFS tree with one level-2 vertex. | Moderately sized tree with branching and various depths. | Larger tree with multiple branches and depth-3 leaf. | No vertices at distance 1, invalid distance sequence |

Table 18: A sample from TAROT dataset comprising 4-tiered test cases: basic, intermediate, complex, and edge. The Reason column details the rationale for each tier assignment.

```
Solve the following coding problem using the programming language python:

The game of Berland poker is played with a deck of $n$ cards, $m$ of which are jokers. $k$
players play this game ($n$ is divisible by $k$).

At the beginning of the game, each player takes $\frac{n}{k}$ cards from the deck (so each card
is taken by exactly one player). The player who has the maximum number of jokers is
the winner, and he gets the number of points equal to $x - y$, where $x$ is the number of
jokers in the winner's hand, and $y$ is the maximum number of jokers among all other
players. If there are two or more players with maximum number of jokers, all of them
are winners and they get 0 points.

Here are some examples: $n = 8$, $m = 3$, $k = 2$. If one player gets 3 jokers and 1 plain
card, and another player gets 0 jokers and 4 plain cards, then the first player is
the winner and gets $3 - 0 = 3$ points; $n = 4$, $m = 2$, $k = 4$. Two players get plain cards,
and the other two players get jokers, so both of them are winners and get 0 points;
$n = 9$, $m = 6$, $k = 3$. If the first player gets 3 jokers, the second player gets 1 joker
and 2 plain cards, and the third player gets 2 jokers and 1 plain card, then the first
player is the winner, and he gets $3 - 2 = 1$ point; $n = 42$, $m = 0$, $k = 7$. Since there are
no jokers, everyone gets 0 jokers, everyone is a winner, and everyone gets 0 points.

Given $n$, $m$ and $k$, calculate the maximum number of points a player can get for winning
the game.

-----Input-----

The first line of the input contains one integer $t$ ($1 \le t \le 500$) | the number of test
cases.

Then the test cases follow. Each test case contains three integers $n$, $m$ and $k$
($2 \le n \le 50$, $0 \le m \le n$, $2 \le k \le n$, $k$ is a divisors of $n$).

-----Output-----

For each test case, print one integer | the maximum number of points a player can get
for winning the game.

-----Example-----
Input
4
8 3 2
4 2 4
9 6 3
42 0 7

Output
3
0
1
0

-----Note-----

Test cases of the example are described in the statement.

The input will be stdin and you should print your solution to stdout

Now solve the problem and return the code.
```

| Test cases | Basic | Intermediate | Complex | Edge |
|---|---|---|---|---|
| **Input** | 3
9 2 3
12 5 4
6 5 3 | 5
20 10 5
15 3 5
10 10 2
14 7 7
18 17 3 | 7
50 25 25
49 49 7
48 20 6
30 0 5
32 16 4
45 23 9
28 14 7 | 6
2 0 2
2 2 2
50 0 25
50 50 50
50 25 5
50 1 2 |
| **Output** | 2
2
0 | 2
3
0
1
0 | 1
0
5
0
5
2
2 | 0
0
0
0
6
1 |
| **Reason** | Simple small cases covering scenarios where jokers are fewer than, equal to, or exceed the per-player limit. | Moderate-sized inputs, testing exact division of jokers, no jokers, and tied maximum distributions. | Varied larger values including big decks, testing heavy distributions and zero-joker scenarios. | Extreme boundary conditions with minimal and maximal n, k, and m values to test edge handling. |