# OpenReview forum: "TAROT: Test-Driven and Capability-Adaptive Curriculum Reinforcement Fine-Tuning for Code Generation"
_ICLR.cc/2026/Conference — ICLR 2026 Conference Withdrawn Submission_

### Official Review · Reviewer_szTy · 2025-10-28

**Soundness:** 2
**Presentation:** 3
**Contribution:** 2
**Rating:** 4
**Confidence:** 3

**Summary:**

TAROT is a novel RFT framework to address the flat and sparse reward landscape of traditional
RFT, which typically relies on a simple binary signal (pass/fail) in code generation tasks. It first
introduces an “intra-problem” curriculum dataset, where each problem is augmented with a fourtier test suite. Based on the dataset, TAROT then applies a curriculum policy with specific reward
weightings, which can dynamically adjust the training focus given the training objective. Extensive
experiment results reveal that the optimal curriculum for RFT is closely tied to the model capability,
with less capable models achieving greater gains with an easy-to-hard progression, whereas more
competent models excel under a hard-first curriculum.

**Strengths:**

1. Well-Motivated RFT Framework: The idea of an "intra-problem" difficulty gradient based on
tiered test cases is a strong and novel contribution. It directly tackles a well-known limitation in
RFT for code generation tasks and is intuitively based on human practices.
2. Significant Empirical Validation: The paper presents extensive experiments across a diverse
set of models (Qwen2.5, Qwen3, Gemma2) and benchmarks (HumanEval, MBPP,
LiveCodeBench, etc.). The results consistently show that the capability-adaptive TAROT
framework significantly outperforms baseline models, demonstrating the method's
effectiveness.
3. Valuable Core Insight: The paper's finding that the optimal curriculum is not one-size-fits-all
but dependent on the model's "effective capability" (a combination of scale and prior
specialization) is a crucial insight for the field. This moves beyond simplistic "easy-to-hard"
curriculum learning.
4. Resource Contribution: The creation and release of the TAROT dataset, with its 15k problems
augmented by 4-tier test suites, is a valuable dataset resource for future research.

**Weaknesses:**

1. Verifiability of the Generated Dataset: my primary concern is the quality and consistency of
the four-tiered test suites. Since they are generated by proprietary LLMs following a promptbased guideline, the actual difficulty distinction between tiers may be inconsistent or difficult to
verify, potentially undermining the core assumption of a structured difficulty gradient.
Additionally, while the authors performed a validation analysis, this reliance on proprietary LLMs
for dataset creation raises questions about reproducibility and scalability.
2. Confounding Factor of Optimization Attribution: The analysis of OOD benchmarks reveals that
the optimal curriculum is also highly task-dependent, not solely a function of model capability.
This finding complicates the central claim, suggesting a more intricate interplay between model
capability, task structure, and curriculum design than the paper currently stated.
3. Weak Correlation of the Reward Signal: The paper intriguingly shows that the final training
reward has only a weak correlation with downstream benchmark performance. This
observation somewhat undermines the centrality of the proposed tiered-reward mechanism. If
the carefully structured reward signal does not reliably predict final performance, its role as the
primary driver of the TAROT framework becomes less clear. It is worth considering whether
downstream tasks could be framed as different (harder or easier) curriculum stages themselves
to be incorporated into the purposed TAROT framework.

**Questions:**

1. The results presented in Figures 3 and 4 consistently show that RFT leads to performance
improvements over the baseline. However, RFT can sometimes lead to performance
degradation or catastrophic forgetting. Did the authors observe any negative results or
instances where a particular curriculum strategy harmed the model's performance, especially
for less-capable models?
2. Could the authors provide a more detailed theoretical or intuitive justification for the key finding
that more competent models excel under a hard-first curriculum? While empirically
demonstrated, a deeper explanation of the underlying learning dynamics would strengthen this
conclusion.

---

> ### Author Response · Authors · 2025-11-25
>
> We thank the reviewer for the clear summary and for engaging deeply with both the strengths and limitations of TAROT.
>
> > 1. Verifiability of the generated dataset and consistency of four-tier suites; reliance on proprietary LLMs.
> >
>
> We share the concern that a tiered dataset must be both *high quality* and *reproducible*. TAROT’s construction addresses these in three ways:
>
> 1. **Ground truth from existing platforms.** We build on coding interview problems that already have a **vetted reference solution and baseline test suites** from production service platforms (Codeforces, CodeChef, HackerRank, etc.,).
> 2. **Black‑box, prompt‑guided tier generation.** We use strong LLMs (OpenAI's o3 and o4) only to generate test cases **subject to strict prompts** that define each tier’s role (Basic/Intermediate/Complex/Edge) and forbid any assumptions about internal implementations.
> 3. **Execution-based validation.** Every generated test case is validated by executing the reference solution provided by the production service platforms. Only tests that pass are retained. This automatically filters out many malformed or semantically incorrect cases.
>
> We then **validate the tiers post hoc** via structural metrics and GPT‑4o re‑labeling (Section 4.2, Figure 2), showing that the assumed difficulty gradient is indeed realized in the data. To further support verifiability, we provide anonymized raw samples ([LINK](https://anonymous.4open.science/r/TAROT-B675/data/data_sample.jsonl)) with multi paged data samples in Appendix I and released the full dataset and generation scripts, enabling the community to regenerate or extend TAROT with different base LLMs if desired.
>
> > 2. Confounding factor of optimization attribution: optimal curriculum is also task-dependent.
> >
>
> We agree and appreciate this nuance. Our results show that the optimal curriculum depends on **both**; the model’s effective capability, and the target task distribution (e.g., HumanEval vs. LiveCodeBench). Our claim is therefore not that “capability alone determines the best curriculum,” but that **capability is a crucial axis that should not be ignored**. In practice, a good curriculum must match *both* the model and the downstream tasks. We soften and clarify the wording around our central claim to reflect this interplay and more explicitly discuss task‑specific effects observed in Section 4.2.
>
> > 3. Weak correlation between training reward and downstream performance; role of the tiered reward.
> >
>
> We agree that this is an intriguing and somewhat sobering observation. In our view it highlights that **tiered reward signal is primarily an *optimization tool***. It shapes the training landscape to avoid flat, sparse rewards and provides richer gradients during reinforcement fine-tuning. However, the resulting *scalar reward value* during training is still **only a proxy** for downstream benchmark performance, especially when those benchmarks differ in format, coverage, or problem style from TAROT.
>
> > 4. Did you observe any negative results or cases where a curriculum harmed performance?
> >
>
> As you suggest, reinforcement fine-tuning is not guaranteed to help, and we intentionally report negative results for transparency.
>
> In particular, Appendix F / Table 6 shows that for Gemma2‑2B‑IT, which is relatively weak on coding tasks, applying Uniform or Complex‑focused curricula can degrade performance below the base model. For example, HumanEval Pass@1 drops from 42.07% (base) to 39.02% (Uniform) and 35.98% (B/I Weighted). We also observe qualitative signs of reward collapse (e.g., repetitive or degenerate outputs) under overly aggressive curricula.
>
> Conversely, a simple **Basic‑only** curriculum improves the same model to 44.51% on HumanEval, illustrating that the *wrong* curriculum can harm performance, while an appropriately “gentle” curriculum can help. These findings strongly support the paper’s core message: curricula must be **capability‑adaptive**; there is no universally safe choice.
>
> > 5. More detailed justification for why stronger models excel under hard‑first curricula.
> >
>
> We see two key learning‑dynamics factors:
>
> 1. **Zone of optimal difficulty.** Strong models already solve Basic and many Intermediate cases with high probability. These tiers provide low‑entropy signals and limited gradient information. Focusing first on Complex/Edge tiers aligns training with the **non‑trivial error frontier**, where gradients are largest and most informative.
>
> 2. **Regularization via later easy cases.** Training solely on hard cases can lead to brittle solutions overfitted to adversarial patterns, which in turn can harm performance on simpler benchmarks. By eventually reintroducing Basic/Intermediate tiers (i.e., complex‑to‑basic schedules rather than “complex only”), TAROT uses easy cases as a **regularizer**, encouraging solutions that are both powerful and robust across the full difficulty spectrum.

---

> > ### Comment · Reviewer_szTy · 2025-11-27
> >
> > While the additional details on execution-based validation are helpful, they do not fully resolve the concern regarding the semantic consistency of the difficulty tiers. Validating that a test case executes against a reference solution ensures correctness, but it does not inherently verify that a "Complex" case is significantly distinct in reasoning difficulty from a "Basic" one. The reliance on proprietary models (o3/o4) to define this gradient creates a "black box" dependency that hampers full reproducibility and transparent analysis of the curriculum's structure.
> >
> > Furthermore, the admission that the optimal curriculum is heavily dependent on the interplay between model capability and specific task distributions suggests that TAROT may lack robustness as a generalizable framework. If the reward signal is an unreliable proxy and the curriculum requires specific tuning for each model-task pair, the practical utility of the framework is somewhat diminished compared to the complexity introduced.
> >
> > Overall, while the core insight of capability-adaptive RFT is valuable, I believe the paper needs further refinement to address these validation and generalization gaps to convincingly support its claims.

---

### Official Review · Reviewer_9Fqg · 2025-10-31

**Soundness:** 3
**Presentation:** 3
**Contribution:** 1
**Rating:** 8
**Confidence:** 4

**Summary:**

TAROT is a test-driven, capability-adaptive framework for RFT of code LLMs. For each problem it builds a four-tier test suite of basic, intermediate, complex, and edge cases while leaving the original prompt and reference solution unchanged. Training uses a tier-weighted objective with two parts: how much training focus to allocate to each tier and how much reward to give each tier. This decouples the curriculum from raw pass rates and lets the same dataset support different learning paths.

Experiments on multiple models and benchmarks show consistent gains and reveal that the best curriculum depends on both model capability and the target task. Weaker models benefit from starting with basic and intermediate cases, while stronger models learn more from emphasizing complex and edge cases. Training rewards rise smoothly but are weak indicators of downstream accuracy, whereas shorter completions correlate better with stronger performance. Overall TAROT improves functional correctness and robustness through intra-problem, test-driven curricula.

**Strengths:**

1. The paper shifts curriculum learning from inter-problem ordering to the intra-problem test gradient and formalizes it with a tier-weighted objective that decouples sampling allocation and reward weighting from raw pass rates, directly addressing flat and imbalanced signals in RFT/RL.

2. The four-tier taxonomy (basic, intermediate, complex, edge) is meaningfully separated through quantitative and qualitative checks, with all tests verified against the reference solution, which reduces noise and supports trustworthy curriculum comparisons.

3. Experiments across multiple models and benchmarks show consistent gains and reveal task-dependent optima; the capability-adaptive guidance is practical for selecting curricula.

4. Community value: The release of a large dataset with validated tiered test suites offers a reusable resource that enables fair, repeatable evaluation of curriculum and RFT/RL methods in code generation.

**Weaknesses:**

The procedure for estimating a model’s effective capability and choosing the corresponding curriculum is under-specified. It remains unclear how to select allocation and weighting for a new model without tuning on target benchmarks, how robust the policy is to different problem orders or tier cardinalities.

**Questions:**

1. How do you estimate a model’s effective capability before training and then select the tier allocation and reward weights for a new model without tuning on the target benchmarks? Please describe any signals you rely on.

2. You generate test cases of different difficulty, but do you have more intrinsic indicators of the actual difficulty achieved? What precise criteria were used to classify test cases into basic, intermediate, complex, and edge tiers during both generation and validation, and how did you verify that these tiers reflect meaningful differences beyond surface features like length or token rarity?

3. In §4.2 you mention pursuing a more holistic measure of intrinsic abilities. How should this be evaluated in practice? Are held-out test sets sufficient, or do you recommend additional diagnostics or probes?

4. Was the problem set and its order kept fixed across all curricula so that only intra-problem tier policies varied? More broadly, do you envision combining a problem-level difficulty curriculum with the test-tier curriculum, and if so, how would you integrate the two while ensuring results are not affected by changes in problem order or by using different problem subsets?

---

> ### Author Response · Authors · 2025-11-25
>
> We thank the reviewer for raising pointed questions on capability estimation, intrinsic difficulty, and curriculum interaction.
>
> > 1. How do you estimate a model’s effective capability before training and then select the tier allocation and reward weights for a new model without tuning on the target benchmarks?
> >
>
> In this work, we rely on **publicly available signals** and a small amount of manual analysis, not tuning on the downstream benchmarks we report. **Public benchmark scores** (e.g., HumanEval, MBPP) provide a coarse but practical indication of whether the model already has strong basic coding and instruction‑following ability. **Model size and specialization** further refine this picture. In our experiments, even smaller code‑specialized models behave more like larger general models in terms of which curricula help them most. Using these signals, we categorize models into “less-capable” vs. “more-capable” and select **Basic‑focused** vs. **Complex/Edge‑weighted** curricula accordingly.
>
> Looking ahead, we believe a more principled and automated approach is both important and enabled by TAROT which we see as future work rather than a solved problem. In the revised paper, we highlight automated curriculum / policy selection as an important direction for future work in the Conclusion:
>
> - **Probe:** Run a small zero-shot evaluation of the target model on TAROT tiers to obtain tier‑wise Pass@1 (Basic, Intermediate, Complex, Edge) and public benchmarks (HumanEval, MBPP, etc.,).
> - **Capability signature:** Aggregate these into a compact “capability signature”.
> - **Policy mapping:** Map this signature to a curriculum policy (e.g., Basic‑focused if Basic is weak, Complex‑focused if Basic is strong but Complex/Edge are weak).
>
> > 2. Intrinsic difficulty indicators for the four tiers and verification beyond surface features.
> >
>
> Tier labels are not assigned post hoc but encoded in the *generation prompts* (Appendix A). For each problem, we instruct the LLMs to produce four test cases; **Basic:** use short, canonical “happy path” inputs to test only core logic, **Intermediate:** add moderate complexity, mixed structures, or typical edge cases, **Complex:** target “advanced functionality and complex logic paths,” often with larger or more intricate inputs designed to break naive strategies, and **Edge:** focus on extreme boundary conditions (e.g., empty inputs, maximum sizes, degenerate cases) to stress robustness.
>
> We then validate that these tiers have **intrinsic difficulty differences** along two axes. **Structural metrics (KDE plots in Figure 2):** input length, token diversity, and character transitions all increase monotonically from Basic to Edge, indicating that harder tiers are not just relabeled variants. **Functional role (Bar chart in Figure 2):** GPT‑4o re‑labeling shows that Complex tiers are predominantly “complexity checks,” and Edge tiers “boundary checks,” confirming that tiers stratify *types* of reasoning and robustness, not just scale.
>
> > 3. In 4.2 you mention a more holistic measure of intrinsic abilities. How should this be evaluated in practice?
> >
>
> We see two complementary directions. **Held-out code benchmarks** remain essential to measure functional correctness and generalization across problem distributions (e.g., HumanEval, MBPP, LiveCodeBench v5). **Targeted diagnostics / probes** can measure finer‑grained abilities that TAROT is designed to exercise: e.g., explicit tests for boundary robustness, algorithmic efficiency, or long‑context reasoning. In practice, we envision using a **combination**: standard benchmarks for overall progress and small TAROT‑style or related probes for diagnosing where a model sits in the intra‑problem difficulty space, which in turn informs curriculum selection as described above.
>
> > 4. Problem set & order: Were they fixed? How to combine problem-level curricula with test-tier curricula?
> >
>
> To isolate the effect of **intra‑problem tier policies**, we **fixed the problem set and its order across all curricula**. All runs use the same 15k problems; only the tier sampling and reward weighting differ. We agree that combining **inter‑problem** and **intra‑problem** curricula is a promising next step. One possible “dual‑axis curriculum” could be that
>
> - inter‑problem axis: adjust sampling probability over problems by an external difficulty estimate.
> - intra‑problem axis: apply TAROT’s tier policies within each sampled problem.

---

### Official Review · Reviewer_VD2u · 2025-10-31

**Soundness:** 2
**Presentation:** 2
**Contribution:** 2
**Rating:** 2
**Confidence:** 4

**Summary:**

This paper introduces a novel technique to generate a curriculum for a LLM learning.

Overall, I am not convinced that the presented work is finished, since the authors provide several techniques to finetune LLM and do not provide a rule to choose one among them.

**Strengths:**

The novel technique to generate data for LLM finetuning.

**Weaknesses:**

The authors in fact propose several techniques. The experiment results suggest that for different models (even among the same family) the most efficient technique is also different. The authors do not propose any strategy to choose one technique.

**Questions:**

Please elaborate on the choice strategy for technique for a model.

---

> ### Author Response · Authors · 2025-11-25
>
> We appreciate the reviewer’s concise summary and for highlighting the missing piece around technique selection.
>
> > 1. The authors propose several techniques. For different models the most efficient technique is different. The authors do not propose any strategy to choose one technique. Please elaborate on the choice strategy.
> >
>
> We agree that, in practice, a method is most useful when it comes with guidance on how to select among its options. TAROT intentionally exposes **multiple curriculum policies** to study their interaction with model capability. From these experiments we derive a **simple, practical heuristic**:
>
> - **Less capable / smaller / non‑specialized models** should use **Basic‑focused or easy‑to‑hard curricula**, which provide dense, reliable reward signals and avoid early reward collapse on hard tiers.
> - **More capable / larger / code‑specialized models** should use **Complex/Edge‑weighted or hard‑first curricula**, which concentrate learning on non‑trivial failures that still challenge the model.
>
> This strategy can be chosen *a priori* based on **Published benchmark performance** (HumanEval, MBPP, etc.), and **Model description** (scale, whether explicitly code‑tuned), without tuning on the target benchmarks used for evaluation in our paper.
>
> Beyond this heuristic, we outline a **future automated framework** (“probe‑then‑select”) where a new model is first evaluated on a small subset of TAROT tiers and public benchmarks to compute a tier‑wise “capability signature,” which is then mapped to a curriculum policy. We view TAROT as providing the empirical foundation for such automated methods rather than claiming to have fully solved the selection problem.
>
> We will clarify this in Section 4.2 and in the Conclusion.

---

### Official Review · Reviewer_njHu · 2025-11-01

**Soundness:** 2
**Presentation:** 1
**Contribution:** 2
**Rating:** 2
**Confidence:** 4

**Summary:**

This paper proposes the TAROT dataset, a Code RL dataset that includes four different difficulty categories (tiers) of test cases, thereby providing more granular/smoother non-binary rewards for Code RL. It also compares several different Curriculum Learning settings. The experimental results show that the proposed RFT method offers some marginal improvement compared to the base model.

**Strengths:**

1. The experiments are extensive. This paper conducts validation on a series of models, including models of different scales and different architectures. It also covers a very comprehensive evaluation set.

2. Overall, I think the core idea of this paper makes sense: providing non-binary rewards to RL through a multi-granularity test case setting and curriculum learning, thereby smoothing the RL optimization process.

**Weaknesses:**

Major concerns:

1.  **Test Case Quality.**  A large number of prior works [1, 2, 3] have demonstrated the importance of test case quality, which encompasses two primary issues: whether a correct solution is incorrectly marked as an error (false negative), and whether an erroneous solution is incorrectly marked as correct (false positive). Based on the construction process of the TAROT dataset, it appears that only a single correct reference solution was used to verify the test case quality. This is clearly insufficient, as it does not guarantee a sufficiently low false positive rate. The false positive issue is also reflected in the limited number of test cases. Without a sufficient quantity of test cases, it is impossible to cover all potential errors. Given that Equation 4 and Table 3 suggest only one test case is included per tier, meaning only four test cases per problem, it raises further doubt as to whether these test cases can effectively screen for incorrect solutions.

2. **The 4 Tiers.** Another key concern is regarding the four tiers proposed in the paper—basic, intermediate, complex, and edge—specifically, whether the corresponding data truly aligns with the definitions on which this paper is based. From Table 4, it appears that the test cases for these four tiers were entirely generated by an LLM, with no other mechanism used to verify, for example, whether the "complex" tier is genuinely more difficult than the "basic" tier. I believe this verification is quite important because the distinction between these tiers is directly related to the effectiveness of the curriculum learning setting. Looking at Table 6, I was unable to observe what kind of differences these curriculum settings exhibited in the end-to-end RL results.

3. **Missing Baseline.** This article indeed features a wealth of experiments, but I still want to point out that a crucial baseline seems to be missing. As far as I know, most current Code RL methods are implemented using the complete test suites, where a full pass yields a $+1$ reward and failing even one test case yields a $-1$ reward. These methods do not use fine-grained reward shaping or curriculum learning (CL). I believe such a naive baseline without CL represents the strategy currently used by most practitioners, and readers would likely be interested in the results for such a baseline.

Minor concerns:

1. I feel that the presentation of this paper needs improvement. Several critical terms, such as 'C/E Weighted,' are not defined within the main text, nor is their full name provided (although I eventually located it in the Appendix). Furthermore, I believe that Table 5 is crucial for understanding the paper and should be moved to a prominent position in the main body. Lastly, it is unclear what the authors intend to convey with Figures 3 and 4. These figures do not compare differences between CL settings, but instead compare the RFT models against a base model. This does not seem aligned with the logic of this paper.

Some typos:

1. L132-133, Duplicated reference for DPO and PPO.

2. L135, The opening quote is a closing quote.

References

[1] Competition-level code generation with alphacode, Science 2022.

[2] Is your code generated by chatgpt really correct? rigorous evaluation of large language models for code generation. NeurIPS 2023.

[3] CodeContests+: High-Quality Test Case Generation for Competitive Programming, EMNLP 2025.

**Questions:**

Please refer to "Weaknesses"

---

> ### Author Response · Authors · 2025-11-25
> **Rebuttal (Part 1/2)**
>
> We thank the reviewer for the clear summary and for highlighting core concerns about test quality, tier validity, and baselines.
>
> > 1. Test Case Quality and limited number of tests per tier.
> >
>
> As noted by the reviewer, test-case quality is critical. TAROT is built on **existing, deployed coding interview problems** with a **single reference solution** that is already vetted by the platforms (Codeforces, CodeChef, HackerRank, etc.,). For each problem, we use the platform’s own test cases as a starting point and then ask a strong LLMs (OpenAI's o3 and o4) to generate four tiers under strict prompts as in **Appendix A, Table 5**. Every generated test case is validated by running the reference solution; only those that pass are kept.
>
> We agree that relying on a single reference solution cannot *eliminate* all false positives/negatives, and that four tests per problem are not sufficient for exhaustive coverage. However, the **scale and diversity** of the dataset substantially mitigate the impact of residual noise, and the goal of this work is to study the *effect of intra-problem structure* on curriculum and reward, not to maximize coverage.
>
> Designing TAROT with exactly one test per tier per problem was a deliberate choice to disentangle “more tests” from “structured tests”; it lets us cleanly analyze how allocation and weighting across tiers affect learning. Extending TAROT to multiple tests per tier is indeed valuable and we consider it a natural next step.
>
> > 2. The 4 Tiers: whether they really match their intended difficulty definitions; lack of verification; Table 6 appears to show small differences.
> >
>
> We share the reviewer’s concern and therefore explicitly validate the tiers along two dimensions. This suggests that tiers differentiate *types* of difficulty, not just surface scale. Further, we added multi pages of concrete raw data samples in Appendix I:
>
> 1. **Quantitative complexity metrics (Figure 2, left).** We do not rely solely on the labels assigned during generation. Instead we compute three structural metrics—input length, token diversity, and character transitions—and show that their distributions shift monotonically from Basic → Intermediate → Complex → Edge. This demonstrates that harder tiers are **not arbitrary** but systematically more complex at the input level.
>
> 2. **Qualitative re‑evaluation (Figure 2, right).** We ask GPT‑4o, as an independent judge, to classify the “role” of each test case (e.g., core logic, complexity stress, boundary robustness). The re‑evaluation shows that Basic and Intermediate tiers focus on “happy path” and moderate complexity, Complex tiers are dominated by algorithmic complexity checks, and Edge tiers overwhelmingly test boundary and corner‑case behavior.
>
> Regarding Table 6 (now Table 8 in the revised menuscript): this table aggregates results across the entire Qwen2.5 family and thus naturally averages away some of the sharp capability‑dependent trends. The capability‑adaptive behavior is clearer in Figure 3, where we show, for example, that Qwen2.5‑1.5B Instruct benefits most from Basic‑focused curricula, while Qwen2.5‑7B Instruct benefits most from Complex/Edge‑weighted curricula. We will make this connection more explicit in the text and clarify that Table 6 is intended for transparency and completeness, whereas Figures 3–4 illustrate the capability-adaptive pattern model‑by‑model.

---

> > ### Author Response · Authors · 2025-11-25
> > **Rebuttal (Part 2/2)**
> >
> > > 3. Missing Baseline: naive full test suite with binary reward and no CL.
> > >
> >
> > We fully agree that this is an important baseline, as it closely reflects current practice in many code RL systems. We therefore **ran two additional baselines** that both use all four tiers per problem as a single, combined test suite but remove any curriculum scheduling or tier-specific reward weighting:
> >
> > - **No Curriculum (Average Rewards):** the reward is the **fraction of tests passed** in the combined suite, i.e., a scaled reward \(R \in [0,1]\).
> > - **No Curriculum (Pass@All):** a **binary** reward, where \(R = 1\) only if all tests in the combined suite pass, and \(R = 0\) otherwise.
> >
> > Due to resource limits, we focus on two representative models (Qwen2.5-1.5B-Instruct and Qwen2.5-7B-Instruct) that span the capability spectrum discussed in the paper. The results, summarized in **Section 4.3**, **Table 3**, show that TAROT consistently outperforms both naïve baselines across all benchmarks, which more cleanly isolates the added value of intra-problem curricula and tiered rewards beyond simply increasing the number of test cases.
> >
> > **Results on Qwen2.5-1.5B-Instruct**:
> > | Setting | HumanEval | HumanEval+ | MBPP | MBPP+ | CodeForces | LiveCodeBench v5 | CruxEval |
> > | --- | --- | --- | --- | --- | --- | --- | --- |
> > | TAROT  | **60.98%** | 55.49% | **51.80%** | **58.20%** | **4.49%** | **5.26%** | **40.20/33.75%** |
> > | Avg. Rewards | 59.15% | 54.27% | 49.20% | 57.93% | 2.72% | 3.70% | 38.75/32.87% |
> > | Pass@All  | **60.98%** | **56.10%** | 44.60% | 53.43% | 2.72% | 3.94% | 34.62/31.75% |
> >
> > **Results on Qwen2.5-7B-Instruct**:
> > | Setting | HumanEval | HumanEval+ | MBPP | MBPP+ | CodeForces | LiveCodeBench v5 | CruxEval |
> > | --- | --- | --- | --- | --- | --- | --- | --- |
> > | TAROT | **84.15%** | **77.44%** | **69.00%** | **70.63%** | **12.95%** | **19.12%** | **57.88/59.38%** |
> > |  Avg. Rewards | 83.75% | 76.22% | 66.00% | 69.84% | 11.41% | 11.95% | 56.37/58.50% |
> > |  Pass@All | 81.10% | 73.78% | 63.00% | 68.52% | 9.49% | 14.81% | 55.62/56.63% |
> >
> > > 4. Minor concerns
> > >
> >
> > We thank the reviewer for these helpful suggestions and reflect them in the revised menuscript:
> >
> > - We defined all abbreviations in the main text and moved Table 5 into a more central position in **Section 4** as **Table 1** to aid comprehension.
> > - We fixed the duplicated reference to DPO/PPO and the minor quotation mark issue.

---

### Official Review · Reviewer_pRbZ · 2025-11-03

**Soundness:** 2
**Presentation:** 3
**Contribution:** 2
**Rating:** 4
**Confidence:** 3

**Summary:**

This paper introduces the TAROT framework for reinforcement fine-tuning in the domain of code generation, which utilizes the varying difficulty of test cases to obtain more training signals. During training, the framework adjusts the proportion and the weight of four tiers of test cases. Experiments demonstrate that this strategy can generally improve the model's performance on coding benchmarks. The authors also observe the diverse training dynamics for base models of different capabilities, which partially explains their setting of curriculum policies.

**Strengths:**

The main strength of this paper is proposing an approach for obtaining more fine-grained rewards by evaluating on test cases of diverse difficulties. This approach may stimulate the model to refine its solutions from general ideas to extreme or corner cases, which differs from the conventional criterion of passing all test cases.

Another thing interesting is the observation that models of different capability have their own favor of training recipes (with respect to test difficulty). It is an analogy of the well-believed idea  that adjusting the distribution of problem difficulty in several training stages.

**Weaknesses:**

Although I generally agree with the authors that it is possible to leverage intra-problem difficulty to to get more reward signals, I think the analysis to the framework is somewhat simplistic. Indeed, the authors observe that in many cases training with TAROT dataset brings about higher score in coding benchmarks, however, the baseline should not be base models themselves. More specifically, as soon as one is able to collect a bunch of data (like the interview dataset used), he can (arbitrarily) generate test cases, perform training with them and expect the possibility of improving the performance. In my opinion, the authors should use the most simple case (may be B/I + average/pass) as the baseline, then the performance gain can be reliably attributed to the weighting or allocation strategies in the proposed framework.

Besides, I am concerned about the quality of different tiers of test cases. Since test cases are generated by LLMs, it is necessary to make sure that they have valid inputs. Additionally, the generated outputs require step-by-step simulation, making it difficult to scale up to stress tests (e.g., n=1e3 or even n=1e5 for sequence length). Also see my comments to Table 3 in the Questions section.

**Questions:**

**A general question:**

Given that there are different training recipes available in the framework, the authors do not provide a method to automatically choose a suitable one for training a specific model (when its capability is unknown to us). Thus, the experimental results are more like iterating over all combinations and select the best one afterwards, which limits the practical applicability of the framework. I understand it is difficult to propose an automatic training framework, thus I would like the authors to share their opinions with respect to this point.



**Question to Figure 1:**

I can understand that it is better to gradually learn harder test cases for less capable models, but what is the insight behind learning from complex to basic test cases for stronger models?


**Question to Line 202-204:**

I generally agree with the exploration of intra-problem difficulty, but I am a bit curious about the definition of "hard" case. In competition-level coding (like CodeForces), small test cases can be passed with brute force, yet large test cases may require algorithms, which filters out undesirable solutions. Does "hard" case aim to examine the correctness of a solution's inner logic, or just an arbitrarily constructed case with larger size compared to "easy" and "intermediate"?



**Question to Table 3:**

Generally I find the contents in the table contain many errors, thus I am concerned about the quality of the dataset as well as the generated test cases.

In the first problem, it says "In each operation, you will be given an integer p and a character c. You have to replace the p-th character of the string with c." However, the test cases do not contain either p or c.

In the second problem, for the intermediate test case "BAABABAB", the answer should be "no", but the test output is "yes".

In the third problem, string S is not provided in the test input.

In the fourth problem, the problem only requires a single input "N", but each test case has two input numbers A and B, and it seems to always output max(A, B).

In the fifth problem, it says "He wants to make the sequence strictly increasing. What is the minimum number of operations he needs to perform?" But the test cases actually correspond to  the opposite case, that is making the sequence strictly decreasing.

In the sixth problem, I personally do not understand the problem description. In the basic test case, how can sequence "1 2 3 4 5" contains some piles of the same maximum height?

Besides, many problems have test cases which contain multiple inner tests (with the first input as the number of inner tests), but is not mentioned in the problem description.



**Minor question and suggestions:**

Line 84: "which is defined via instruction-following fidelity and baseline coding proficiency", but the authors do not explain these two ideas in the paper. I think this part is confusing and can be deleted.

Line 135 and afterwards: The left quotation marks look weird. In LaTeX, left and right quotation marks are typed differently.

Line 271 (Figure 2): The authors should explain the meaning and unit of x-axis in Figure 2. How are token diversity and transitions calculated?

Line 289: The citation for the dataset is missing.

Line 290-292: The authors need to explain the different strategies (Forward/Reversed/Static and Uniform, B&I, C&E).

Line 299: It is necessary to mention some important details of implementation here, such as the RFT algorithm used. Do not put everything in the Appendix.

Line 430: "However, the reward observed during training does not reliably anticipate downstream benchmark outcomes." I suggest conducting a case study to understand this phenomenon. That is, given that these models are already capable of writing codes (also demonstrated by baseline performance in benchmarks), why do they gain such low reward in the beginning shown in Figure 5(a)？

Line 432 (Figure 5): Please briefly explain what the light-colored trajectories are. Also, please explain why the trajectories of some models stopped in the middle (it is confusing, although I get to know the reason from Appendix).

Line 865 (Table 5): When not all tiers are used, is the loss function computed with relative weight (of available tiers)?

---

> ### Author Response · Authors · 2025-11-25
> **Rebuttal (Part 1/2)**
>
> We thank the reviewer for the thoughtful analysis and for highlighting both conceptual and practical issues.
>
> > 1. The baseline should not be base models themselves. A simpler intra‑problem baseline (e.g., B/I + average/pass) would better isolate the contribution of weighting and allocation. Besides, I am concerned about the quality of different tiers of test cases…
> >
>
> **Baseline choice and attribution of gains.** We agree that comparing only against the base checkpoints is not sufficient to attribute gains specifically to the proposed curriculum and weighting strategies. In addition to our existing comparisons among different TAROT curricula, we ran a more “naïve” yet realistic baseline that uses all four tiers but no curriculum or reward shaping. Current revision, now detailed in the **"Comparison with Standard Reward Schemes"** section **4.3, Table 3**, show that TAROT consistently outperforms both an **Avg‑reward baseline** and a strict **Pass@All baseline** that only rewards solutions passing all tiers.
>
> **Tier quality, validity, and scalability.** All TAROT test cases are generated from **existing, verified interview problems** with a known reference solution from a deployed coding interview platforms (Codeforces, Codechef, HackerRank, etc.,). As described in Appendix A, we seed the prompts with the platform’s original test cases and problem statements and then ask a strong LLMs (OpenAI's o3 and o4) to generate Basic/Intermediate/Complex/Edge variants under clear black‑box constraints (no access to the solution code). Every generated test case is then **validated by executing the reference solution**; only cases that pass are retained.
>
> We agree that (i) some false positives/negatives are inevitable at this scale, and (ii) more test cases per tier would generally improve coverage. For this first study, however, we deliberately restrict to **one test case per tier** to isolate the effect of intra‑problem structure on curriculum and reward, rather than maximizing raw coverage. Scaling TAROT to multiple tests per tier, and comparing coverage vs. learning dynamics, is a natural extension we plan to explore in follow‑up work.
>
> > 2. A general question: … no method to automatically choose a suitable training recipe when model capability is unknown. Results look like iterating combinations and picking the best afterwards.
>
> We appreciate this important point. Our goal in this paper is to **demonstrate** that intra‑problem curricula should depend on model capability, not to fully solve automated curriculum selection. That said, our experiments do yield **actionable guidance**. **Weaker or smaller, non-specialized models** benefit most from Basic‑focused or easy‑to‑hard curricula, which prevent reward collapse on hard tiers. **Stronger or code‑specialized models** gain more from Complex/Edge‑focused or hard‑first curricula, which concentrate learning on their “zone of optimal difficulty.”
>
> These choices can already be made *without* tuning on target benchmarks, using publicly available signals:
>
> 1. **Public benchmark scores** (HumanEval, MBPP, etc.) and instruction‑following metrics indicate whether a model is already strong on basic coding tasks.
> 2. **Model size and specialization**: even at smaller scales, code‑specialized models exhibit learning patterns similar to larger general models, suggesting that “strong vs. weak” is better captured by *effective coding capability* than by parameter count alone.
>
> We also outline a rough direction toward a **fully automatic framework**, which we see as future work rather than a solved problem. In the revised paper, we highlight automated curriculum / policy selection as an important direction for future work in the Conclusion:
>
> - **Probe:** Run a small zero-shot evaluation of the target model on TAROT tiers to obtain tier‑wise Pass@1 (Basic, Intermediate, Complex, Edge) and public benchmarks (HumanEval, MBPP, etc.,).
> - **Capability signature:** Aggregate these into a compact “capability signature”.
> - **Policy mapping:** Map this signature to a curriculum policy (e.g., Basic‑focused if Basic is weak, Complex‑focused if Basic is strong but Complex/Edge are weak).

---

> ### Author Response · Authors · 2025-11-25
> **Rebuttal (Part 2/2)**
>
> > 3. Question to Figure 1: … what is the insight behind learning from complex to basic test cases for stronger models?
> >
>
> For stronger models, the main intuition is to **concentrate learning signal in their “zone of optimal difficulty”** while avoiding overfitting to only the hardest tier. Strong models already solve many Basic/Intermediate cases with high probability. Rewarding them heavily for these trivial successes **wastes capacity** and can even introduce noise if they start exploring unnecessarily complex solutions to easy tests. Focusing early on Complex/Edge tiers emphasizes cases that still elicit genuine errors and thus provide **informative gradients**.
>
> However, training exclusively on Complex/Edge tiers for too long risks **overfitting** to these stress cases and degrading performance on simpler ones (e.g., solving Basic cases with over-complicated logic that is brittle to small variations). Hence our “hard‑first, then broaden” curricula: we begin with complex tiers to push the frontier, then gradually reintroduce Basic/Intermediate tiers to “flatten” the learning signal and regularize behavior.
>
> In the revised paper, we explicitly added a paragraph about “zone of optimal difficulty” in **Section 4.2**.
>
> > 4. Question on the definition of “hard” cases and Codeforces analogy.
> >
>
> As clarified in Appendix A (Table 5), all tiers are generated under a black‑box testing principle: the prompt explicitly forbids any assumptions about internal implementations and specifies the roles of the basic, intermediate, complex, and edge tests since the target model’s solution is unknown at generation time.
>
> The tiers are not arbitrary. Prompts instruct the LLM to:
>
> - **Basic:** focus on short, “happy path” cases that test core logic under simple inputs.
> - **Intermediate:** introduce mixed types, moderate edge cases, and slightly larger inputs.
> - **Complex:** stress “advanced functionality and complex logic paths,” often via larger, more structured inputs that filter out naive or brute-force strategies (analogous to using larger (N) on Codeforces to reject (O(N^2)) solutions).
> - **Edge:** target extreme boundary conditions and special cases (empty arrays, maximum values, degenerate structures, etc.) to reveal hidden flaws in inner logic.
>
> As depicted in **Section 4.2**, **Figure 2**, we then confirm that this intent materializes in the data via both **quantitative metrics** (longer, more diverse, higher‑transition inputs for harder tiers) and a **qualitative GPT‑4o re‑evaluation** that labels Complex cases as primarily testing complexity and Edge cases as primarily testing boundary robustness.
>
> > 5. Question to Table 3 and concerns about errors and dataset quality.
> >
>
> We thank the reviewer for carefully reading Table 3 (now removed in the revision) and identifying multiple inconsistencies. You are absolutely right: in the process of compressing several problems onto a single page, some examples were manually edited and then inadvertently corrupted when copying and pasting. We sincerely apologize for the confusion this caused. Crucially, these errors affect only the rendered illustrative examples in Table 3, *not* the underlying TAROT dataset used for training and evaluation. As described in Appendix B, all dataset entries are generated and stored in a fully automated pipeline.
>
> To restore trust and demonstrate the true structure and quality of the dataset, we have included multiple pages of raw samples in the Appendix I, showing exactly what the models see during training. Also, you can find the the raw JSONL file ([LINK](https://anonymous.4open.science/r/TAROT-B675/data/data_sample.jsonl)) containing unedited examples (problem statement + four tiers + generation rationales).
>
> > 6. Minor questions and suggestions
> >
>
> We appreciate these detailed suggestions and have addressed them comprehensively in the revised manuscript including:
>
> - **L290–299 / Moving Definitions to Core Section**: We have moved the definitions of our core curriculum strategies (Forward, Reversed, Static) and reward weighting schemes (Uniform, B&I, C&E), as well as the GRPO specification, from the appendix into Section 4.1 for greater clarity and accessibility.
>
> - **Figure 2 (L271) / Defining Diversity Metrics**: We have clarified the visualization in Figure 2 by defining precisely how Token Diversity (unique/total token ratio) and Transitions (frequency of character-class switches) are computed, and explicitly stated that the x-axis shows the density over each metric.
>
> - **Figure 5 (L432) / Interpreting Trajectories**: We now explicitly state that semi-transparent lines represent alternative curriculum strategies, while the solid, annotated lines correspond to the best-performing strategy for each model, and explain that some trajectories terminate earlier due to the different batch/accumulation settings used for various model sizes under a fixed compute budget.

---

### Official Review · Reviewer_sG53 · 2025-11-05

**Soundness:** 2
**Presentation:** 2
**Contribution:** 1
**Rating:** 2
**Confidence:** 5

**Summary:**

This paper proposes TAROT, a test-driven and capability-adaptive curriculum reinforcement fine-tuning framework that trains code generation models through multi-level test rewards and ability-based curriculum scheduling to improve correctness, reasoning, and robustness.

**Strengths:**

The paper includes curriculum learning into code generation of RL domain.

**Weaknesses:**

1. The weighting scheme for small/medium/large models is purely empirical, and the “difficulty” definition of tests is also rule-based without ablation studies to justify its validity. This makes the method feel ad-hoc and lowers novelty.

2. The paper only evaluates PPO. Given the rule-based reward (pass rate), GRPO or similar gradient-regularized policy optimization methods seem more appropriate and potentially stronger. Not testing them weakens the experimental rigor.


3. Figures and tables are not well organized — too many figures per page and captions placed inconsistently (tables should have captions below). The improvement numbers are hard to locate and interpret.


4. The evaluated models are already outdated for ICLR 2026 standards, and there are no comparisons against newer or closed-source models. This limits the relevance of the conclusions.

5. Lack of novelty. Just empirical reward shaping under the situation that is the baseline is not convincing.

**Questions:**

Why does this paper use PPO, instead of GRPO? Cause code is verifiable so that GRPO is straightforward.

---

> ### Author Response · Authors · 2025-11-25
> **Rebuttal (Part 1/2)**
>
> We thank the reviewer for the detailed and constructive feedback.
>
> > 1. The weighting scheme for small/medium/large models is purely empirical, and the “difficulty” definition of tests is also rule-based without ablation studies to justify its validity. This makes the method feel ad-hoc and lowers novelty.
> >
>
> We agree that the weighting scheme is empirical. However, it is not arbitrary: it is grounded in structural analyses of the generated tiers rather than manual intuition.
>
> First, we validate the “difficulty” definition using **three quantitative metrics** and **one qualitative judge**, as shown in Figure 2. The KDE plots demonstrate a consistent rightward shift from Basic → Intermediate → Complex → Edge in (i) input length, (ii) token diversity, and (iii) character transitions, indicating that the harder tiers are structurally more complex, not just relabeled variants. In addition, GPT‑4o re-evaluation of the tiers confirms that Complex cases are predominantly judged as stressing algorithmic complexity, whereas Edge cases are judged as targeting boundary and robustness checks.
>
> Second, the core novelty of TAROT is not the specific numerical values of the weights, but the **intra-problem difficulty hierarchy** and the **decoupling of tier allocation from reward weighting**. We show that (i) intra-problem tiers provide a strictly richer reward landscape than a flat pass/fail signal, and (ii) the optimal allocation over these tiers depends systematically on model capability. This “capability-adaptive,” intra-problem curriculum has, to our knowledge, not been explored in prior code RL work.
>
> We agree that explicit ablations over alternative weighting schemes (e.g., more aggressive or flatter weights) would further clarify sensitivity. In this revision, we partially address this by (i) **comparing multiple reward-weighting templates** (Uniform, B/I-weighted, C/E-weighted, Basic-only, Complex-only) in 4.1, Table 1, and (ii) **adding Comparison with Standard Reward Schemes** section that contrasts TAROT with Avg-reward and Pass@All baselines in 4.3. A more exhaustive sweep over more extreme weight configurations remains an interesting direction for future work.
>
> > 2. Why does this paper use PPO, instead of GRPO? Cause code is verifiable so that GRPO is straightforward.
> >
>
> We apologize for the confusion in the current draft. Our training framework **already uses GRPO**, not vanilla PPO. **We adopted a GRPO objective** (as detailed in the original Section 4.1 and Appendix B) because, as the reviewer notes, verifiable code rewards (pass rate) are naturally suited to gradient-regularized policy optimization. We also reported an ablation on the GRPO hyperparameter ($\beta$) (Appendix D and Figure 5) and analyzed the impact of the sampling temperature (Figure 6).
>
> **In the revised version, we have clarified this point significantly**: We now explicitly state in the Introduction and Section 4.1 that all RFT experiments utilize GRPO. Furthermore, we clearly separate the algorithm choice (GRPO) from the curriculum/reward contributions of TAROT, which are orthogonal and could, in principle, be combined with other policy optimization methods.
>
> > 3. Figures and tables are not well organized — too many figures per page and captions placed inconsistently (tables should have captions below). The improvement numbers are hard to locate and interpret.
> >
>
> We appreciate this comment on presentation. We strictly followed the official ICLR LaTeX template, which places **table captions above** tables and **figure captions below** figures; all captions are consistent with that convention. That said, we agree that some pages are visually dense. In a revised version, we made the following changes:
> - reduce the number of figures per page where possible
> - make the improvement numbers more visually salient (e.g., bolding deltas, adding arrows)

---

> > ### Author Response · Authors · 2025-11-25
> > **Rebuttal (Part 2/2)**
> >
> > > 4. The evaluated models are already outdated for ICLR 2026 standards, and there are no comparisons against newer or closed-source models. This limits the relevance of the conclusions.
> > >
> >
> > We respectfully disagree that the models are outdated relative to the ICLR 2026 timeline. Our experiments include **Gemma 2** (released June 2024), **Qwen2.5** (released September 2024), and **Qwen3‑4B‑Instruct** (released April 2025) covering both general-purpose and code-specialized variants across multiple sizes. These represent the frontier of open-weights models around the submission deadline and, importantly, span a meaningful range of capabilities. Our goal in TAROT is to study **capability-adaptive curricula** under a controlled, open-weight setting rather than to chase the absolute best closed models. We will clarify this scope and positioning in the Introduction and Discussion.
> >
> > > 5. Lack of novelty. Just empirical reward shaping under the situation that is the baseline is not convincing.
> > >
> >
> > We appreciate the concern about novelty and agree that simple reward tweaking alone would not be sufficient. TAROT’s contribution goes beyond “reward shaping” in two ways:
> >
> > 1. **Intra-problem granularity.** Prior curriculum work for code mostly focuses on **inter‑problem** ordering (e.g., reordering or filtering entire problems). TAROT instead constructs a *four‑tier test suite per problem* (Basic, Intermediate, Complex, Edge). This is closer to Test‑Driven Development practice and directly addresses the flat reward landscape by injecting a structured intra‑problem difficulty gradient.
> >
> > 2. **Decoupled curriculum design.** TAROT explicitly decouples, **allocation**: how frequently each tier is sampled during training, from **reward weights**: how much reward each tier contributes when passed.
> >
> > This separation allows the *same dataset* to support different learning paths depending on model capability. Our experiments show that weaker models benefit from Basic‑focused, easy‑to‑hard curricula, whereas stronger or code‑specialized models benefit from Complex/Edge‑focused curricula.
> >
> > Together with the release of a large, validated intra‑problem dataset ([LINK](https://anonymous.4open.science/r/TAROT-B675/data/data_sample.jsonl)), we view TAROT as introducing a **new axis for curriculum design in code RL**, rather than just tuning a single reward function on a fixed test suite.

---

### Comment · Area_Chair_3emr · 2025-11-25
**Author Responses Are Ready - Please Review & Provide Feedback**

Dear Reviewers,

Thank you once again for your essential contributions to the review process. The authors have submitted their responses to your initial reviews.

I kindly ask you to carefully review the authors' responses for the papers you are handling. Your timely assessment of how the authors have addressed your original concerns is a critical step in reaching a final decision.

Please provide your feedback and any necessary updates to your reviews as soon as possible to ensure we can meet our tight schedule for the discussion phase.

Your prompt attention to this matter is highly appreciated.

Best regards,
Area Chair

---

### Note · Authors · 2026-01-05

I have read and agree with the venue's withdrawal policy on behalf of myself and my co-authors.